# Persistent warm-eddy transport to Antarctic ice shelves driven by enhanced summer westerlies

Libao Gao [1,2,3] ✉, Xiaojun Yuan [4], Wenju Cai [2,5,6] ✉, Guijun Guo [1,2,3], Weidong Yu[2,7], Jiuxin Shi [2,6], Fangli Qiao [1,2,3], Zexun Wei [1,2,3] & Guy D. Williams[1]

The offshore ocean heat supplied to the Antarctic continental shelves by warm eddies has the potential to greatly impact the melting rates of ice shelves and subsequent global sea level rise. While featured in modeling and some observational studies, the processes around how these warm eddies form and overcome the dynamic sub-surface barrier of the Antarctic Slope Front over the upper continental slope has not yet been clarified. Here we report on the detailed observations of persistent eddies carrying warm modified Circumpolar Deep Water (CDW) onto the continental shelf of Prydz Bay, East Antarctica, using subsurface mooring and hydrographic section data from 2013-2015. We show the warm-eddy transport is most active when the summer westerlies strengthen, which promotes the upwelling of CDW and initiates eddy formation and intrusions. Our study highlights the important role of warm eddies in the melting of Antarctica's ice shelves, both now and into the future.

The Southern Ocean plays a critical role in heat uptake and redistribution in the context of global warming in recent decades[1–4]. Warm intrusions of modified Circumpolar Deep Water (mCDW), the most important southward oceanic heat transport to the Antarctic margin, have the potential to greatly increase the supply of oceanic heat to the cavities of Antarctic ice shelves, and therefore enhance basal melting and its subsequent contribution to global sea-level rise[5–18]. These intrusions are sourced from warm, saline and nutrient-rich CDW, at depth in the eastward-flowing Antarctic Circumpolar Current (ACC).

To reach the ice shelves, the CDW needs to migrate poleward, upwell over the upper continental slope to the depth range of the continental shelf break and overcome the dynamical barrier of the Antarctic Slope Front (ASF). The ASF is a dynamic sub-surface barrier between the cold shelf water and the warmer offshore water around Antarctica, which inhibits the transport of mCDW onto most of the continental shelves[19–21]. It is important to note that the characteristics of the continental shelf regions surrounding East Antarctica differ significantly from the West Antarctic regions. In the western areas, there are primarily warm shelves (bathed in waters with temperatures exceeding 0 °C), and the ASF is either weak or absent[21]. Conversely, in the eastern regions, the waters on the shelves are cold and dense, and the ASF is well-defined[21]. The ASF is often associated with the westward Antarctic Slope Current (ASC) and the easterly Antarctic coastal wind regime[20,21]. Thereafter, mCDW intrusions need to negotiate the shelf break and move across the continental shelf to penetrate beneath Antarctic ice shelves. The lower the 'modification' of the intrusion, the greater the oceanic heat supply for potential basal melting.

Many hydrographic observations have captured the warm water intrusions onto the continental shelves around Antarctica, including the Prydz Bay[22,23], Adélie Land[24], and the Amundsen and Bellingshausen

[1]First Institute of Oceanography, Key Laboratory of Marine Science and Numerical Modeling, Ministry of Natural Resources, Qingdao, China. [2]Laboratory for Regional Oceanography and Numerical Modeling, Laoshan Laboratory, Qingdao, China. [3]Shandong Key Laboratory of Marine Science and Numerical Modeling, Qingdao, China. [4]Lamont-Doherty Earth Observatory, Columbia University, New York, USA. [5]Centre for Southern Hemisphere Oceans Research, CSIRO Oceans & Atmosphere, Hobart, Australia. [6]Physical Oceanography Laboratory, Ocean University of China, Qingdao, China. [7]School of Atmospheric Sciences, Sun Yat-Sen University, Zhuhai, China. ✉e-mail: gaolb@fio.org.cn; wenju.cai@csiro.au

Seas[9,11,25–27]. Concurrently, several modeling studies have successfully simulated CDW intrusions and elucidated their role in ocean heat transport to the continental shelf and ice shelves[6,13,28–32]. Eddy-resolving models indicate that poleward mesoscale eddies, which have emerged due to the intensified mid-latitude westerly winds and their concurrent poleward shift over the past 50 years[33–35], play a crucial role in the Antarctic overturning circulation. These eddies are a significant factor in overcoming the barrier imposed by the ASF and enabling an enhanced transfer of relatively warm mCDW towards the Antarctic ice shelves[36].

And yet despite this poleward intensification of the westerly winds, both hydrographic observations and climate models show little change in Southern Ocean stratification or net ACC transport over the recent decades. This lack of change is attributed to an enhanced poleward eddy transport[37–40], after several modeling studies proposed that stronger eddy fluxes could compensate for the larger northward Ekman flow caused by the additional input of westerly momentum[41–46].

Several observational studies have described cross-slope eddies near the Antarctic continental shelves[47–49]. However, precisely how eddies traverse the ASF barrier to facilitate the transfer of relatively warm waters to the shelf regions remains poorly understood. Here we report on the detailed observations of persistent warm-eddy transport to an Antarctic ice shelf from a subsurface mooring and ancillary data from hydrographic sections and instrumented seals. We link the processes of warm-eddy formation and transport to enhanced westerly winds in summer and speculate on the future of these processes should the westerlies strengthen further as predicted.

## Results

### Observations of warm eddies onto the continental shelf

Oceanographic Conductivity-Temperature-Depth (CTD) sections, subsurface mooring and elephant seal CTD data were collected in the Prydz Bay region of East Antarctica across 2013–2015 (Fig. 1a). There are six meridional hydrological sections across the continental slope and shelf with 2.5 degrees spacing in longitude, one zonal transect across the Prydz Channel along 67.25ºS and one additional transect along the Amery Ice Shelf. A subsurface mooring located in the Prydz Channel (72.11ºE, 67.18ºS) was deployed in March 2013 and recovered in February 2015. In addition, there are numerous seal CTD profiles available in this region, with many concentrated in and around the coastal polynya areas[23,50].

The potential temperature and salinity data of the 73ºE section and zonal section are presented in Fig. 1b–e. An inflow of relatively warm (>-1.7 °C) and salty (34.4-34.6) water (hereafter defined as mCDW) intrudes onto the continental shelf during the observational period in February 2013 (Fig. 1b, d). The mCDW reaches up to 100 m depth and extends southward to 67.8ºS along the 73ºE section, mixing with the relatively cold shelf waters (Fig. 1b). The main intrusion of mCDW across the zonal section occurred in the 200-400 m layer between 72ºE-76ºE. The maxima in potential temperature (−0.2 °C) and salinity (34.6) are found at 73.3ºE at a depth of 350 m (Fig. 1d). In February 2015, the observed mCDW intrusion seemed absent (Fig. 1c, e). The inflowing mCDW was constrained below 250 m on the continental shelf break north of 67ºS. The distribution of temperature and salinity across this section over the two years shows the substantial temporal and spatial variability of mCDW intrusions in this region.

The two year-round subsurface mooring in the Prydz Channel enables further understanding of the temporal variability of the mCDW intrusions (Fig. 2). Hourly potential temperature records at the five depth levels (248 m, 270 m, 325 m, 449 m, and 520 m) of the mooring show a clear seasonal cycle (Fig. 2a, Supplementary Figs. 1 and 2). Warm water (with maxima over −1 °C) appears during the austral summer (November-April), and the signal is intensified in the upper three levels. The warmest record

(−0.26 °C) occurred at 325 m in April 2014. During the austral winter (May-October), Winter Water (WW) reached the bottom layer and the potential temperatures at all depths were near −1.89 °C with little variation, suggesting strong mixing due to deep convection across the sea-ice growth season.

Following previous studies[7,25], we identify 28 eddy-like warm events from the two-year records of temperature and current velocity at 270 m (Fig. 2b). All warm events occurred during the austral summer season, with the highest frequency of events in April 2014 and January 2015. The cross-stream velocity $v_e$ (perpendicular to the background velocity, see Materials and Methods) is a suitable parameter to identify the characteristic signature of eddies, as initially positive (negative) velocities are indicative of anticyclonic (cyclonic) circulation (Fig. 3a, b). The cross-stream velocity derived from the current meter at 270 m (Fig. 2b) shows that the amplitude is 3-8 cm/s during the entire time series compared to the mean background velocity (2.4 ± 0.3 cm/s). Unlike the seasonal cycle of potential temperature, the cross-stream velocity varies throughout the year, likely reflecting an intra-seasonal oscillation. This distinction will be explained in more details later.

The potential temperature signature and associated heat flux are calculated to estimate the thermal effect of the warm events (Fig. 2c, d). Figure 2d shows that while most of the warm signals are above 450 m, some can reach the seafloor. The warmest event occurred in April 2014. Most of the warm events (~ 75%) resulted in a positive heat flux (between 50-100 w/m$^2$) southward (positive values) into the Prydz Bay basin, although the remaining warm events (~ 25%) removed heat (< 50 w/m$^2$) from the basin (negative values). Analysis of the background velocity at 270 m suggests that the warm events propagating out of the basin are not the same well-mixed warm eddies retreating back to the continental shelf break (Supplementary Fig. 3). The strongest heat fluxes (> 150 w/m$^2$) occurred in January 2015. The warmest signal (in April 2014) shows a relatively small heat flux (nearly 50 w/m$^2$), resulting from the weak background velocity. Assuming that all of the eddy heat is available for basal melting of the area beneath the Amery Ice Shelf that the mCDW can potentially access (about 20% of the total area), the poleward heat flux associated with the eddy intrusions (0.79 ± 0.15 Sv) would result in an average basal melt rate of 3.1 ± 0.7 m of ice per year over this section of the ice shelf (see Methods). This is roughly 50% more than the previous estimate of annual basal melt rate (2.0 ± 0.5 m/year) in the same area driven by mCDW flux into the Amery Ice Shelf cavity[16]. Despite the heat lost over the continental shelf through interaction with polynyas[50], some mCDW can still reach the Amery ice shelf[16,51]. This suggests that the eddy-induced heat transport into Prydz Bay has an important impact on the mass balance of the Amery ice shelf.

### Characteristics and evolution of the warm-eddy intrusions

Composite analysis of the warm events helps understand the structure and process of the eddy intrusions (Fig. 3). Importantly, all 28 warm events are strongly related to eddy rotation. The cross-stream velocity $v_e$ at 270 m determines whether an eddy is cyclonic (17 events) or anticyclonic (11 events). The amplitude of the cross-stream velocity signal in the cyclonic events varies between 1-5 cm/s, with an average peak value of less than 2.8 cm/s. The current velocities of the anticyclonic events have comparable amplitudes (2.5 cm/s in average) to those of the cyclonic events, but with the opposite phase (Fig. 3a–c). Assembling composites of the temperature evolution during the eddy period (Fig. 3d–f) provides the associated eddy structure, which is inherently difficult to observe directly. On average, the cyclonic events occur above 450 m with a warm core (>-1.4 °C) located above 300 m. The anticyclonic events are also concentrated in the same layer, but in contrast, the warm core (>-1.4 °C) is deeper at a depth of 350 m. The

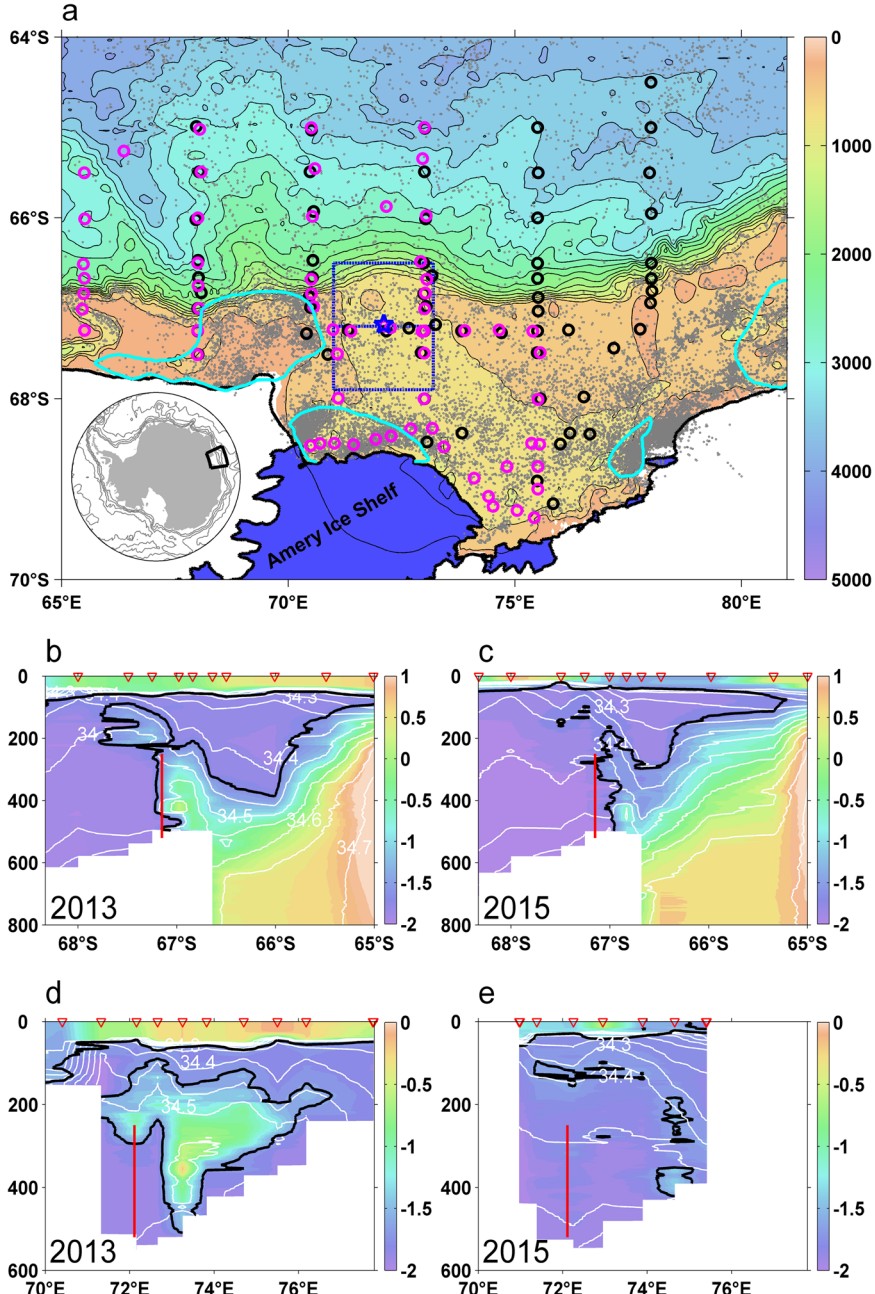

**Fig. 1 | Study area, data locations and warm water intrusions into Prydz Bay during 2013-2015. a** Bathymetric map. Circles show Conductivity-Temperature-Depth (CTD) stations occupied during 2013 (black) and 2015 (purple), gray dots show elephant seal data locations, the mooring location is shown with a blue star, and blue boxes cover the Prydz Channel. Polynyas are shown as cyan lines[23]. (**b**–**e**)

Potential temperature (shade) and salinity (white contours) from CTD transects in 2013 and 2015. Black line indicates −1.7 °C contour. Red line represents subsurface mooring. (**b**, **c**) show the data along 73°E transect. (**d**, **e**) show the data from the zonal transects over the Prydz Channel near 67.25ºS.

vertical structure in temperature (Fig. 3g–i) shows that the anticyclonic events are warmer than the cyclonic events across the full-depth. The maximum difference is around 0.13 °C over the 270-320 m layer. In general, anticyclonic rotation results in convergence and downwelling, and may be responsible for the relatively warm eddies observed. The opposite is the case for the cyclonic eddies.

The monthly distribution of warm-eddy intrusions reveals that the warm events occur across the October-April season (1.2 events per month on average), consistent with Fig. 2b. The maximum number of cyclonic (anticyclonic) events occurs in January (April – see Fig. 4a). The duration of these events, as they pass the mooring location, ranges from 1–7 days, most of which are within 2-3 days (78.6%, 22 of 28

events). A total of 63% of the anticyclonic (7 out of 11) events take 3 days to pass the mooring area (Fig. 4b).

As it is unlikely that many eddies pass directly over the mooring, the horizontal scale of the eddy chords can be estimated from the duration and background velocity. Figure 4d indicates that the eddy chords vary within 13.5 km as a decreasing function. Events with a horizontal scale of 0-5 km, 5-10 km and 10-15 km account for 71.4%, 21.4% and 7.2% of the data, respectively. Compared with the mesoscale eddies in the tropical ocean, the horizontal scales of eddies in polar region are relatively small, associated with the smaller Rossby deformation radius (3-4 km) at higher latitudes. Based on the potential temperature from the mooring records (Fig. 4c), about 11.2% (20.4%) of

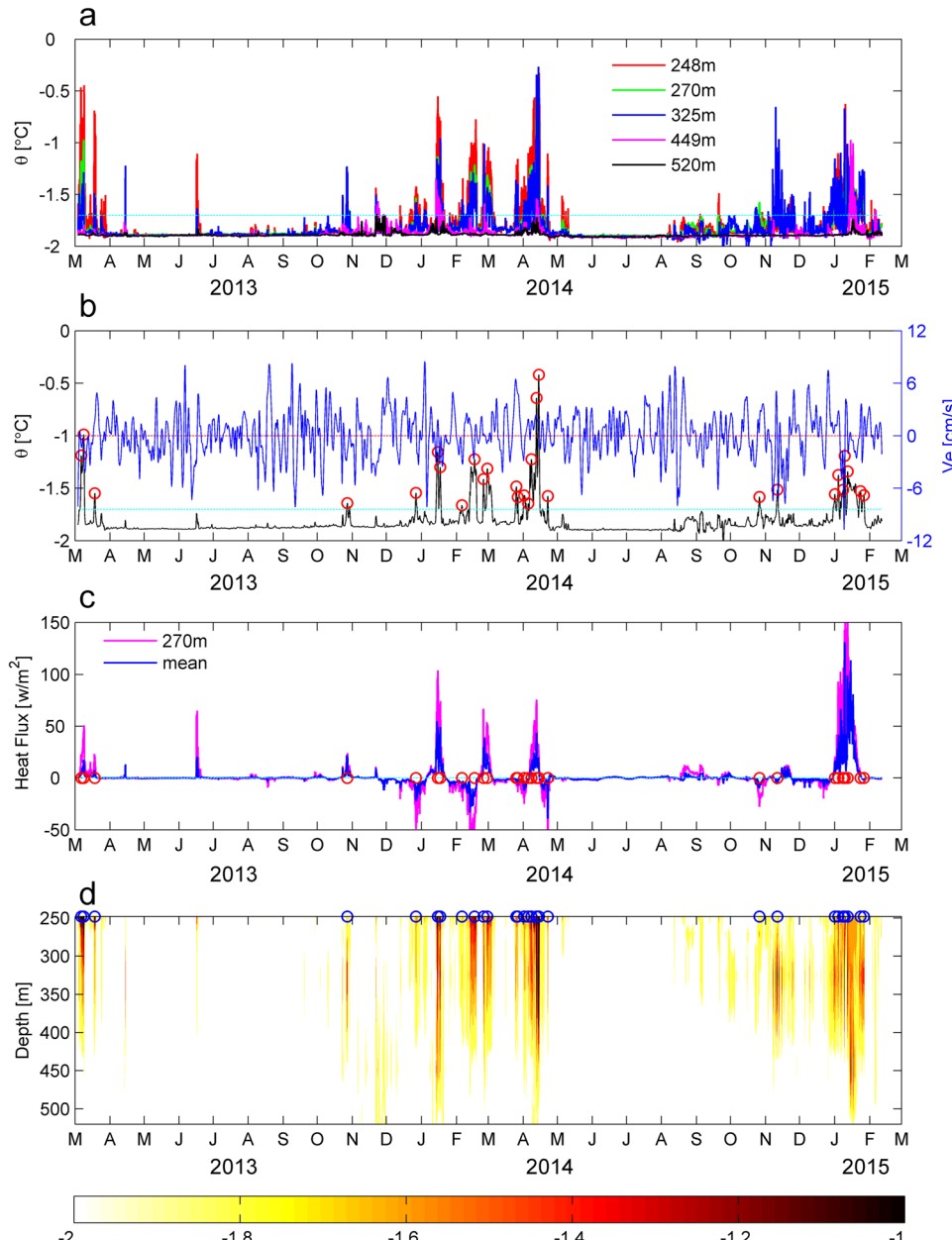

**Fig. 2 | Observed warm intrusion events at mooring location during 2013-2015.**
**a** Hourly potential temperature at different depth levels (248 m, 270 m, 325 m, 449 m, and 520 m). **b** Hourly potential temperature and cross-stream velocity records ($v_e$) at 270 m. **c** Depth-integrated (blue) and 270 m level (purple) heat fluxes. **d** Interpolated potential temperature evolution from the mooring sensors. Cyan lines in **a** and **b** show −1.7 °C position. Isolated warm events are marked with circles in (**b–d**).

the temperature records are warmer than -1.7 °C (-1.8 °C). This indicates there is the equivalent of 1.4-2.4 months of warm-eddy intrusions every year.

It is worth noting that the eddy chords estimated from the mooring record are around 10% ~ 50% of the full eddy size observed from the altimetry data. There are four possible explanations for this. Firstly, the mooring record used to estimate the eddy chords is observed by a current meter located at 270 m (Figs. 2b and 3). Since the three-dimensional shape of eddies are vertical-taper structures (Fig. 3d–f), the eddy chords at 270 m must be much smaller than the length scales at the sea surface. Secondly, the majority of the eddy chords do not cross through the center of the eddies (assumed to be circular in the horizontal plane). Thirdly, the mooring is located at 67.18ºS, approximately 100 km from the continental shelf break. Intruding eddies could move eastward with the Prydz Bay Gyre,

preventing the entire eddy from passing over the mooring location. Finally, since the merged Sea Surface Height (SSH) data is provided on a 0.25° grid, the smaller eddies just are not resolved by the altimetry product. All four reasons explain the reduction in eddy chord lengths estimated from the mooring record.

While the warm events only occur during austral summer, the cross-stream velocity $v_e$ varies throughout the year (Fig. 2b and Fig. 4a). To explain this difference, we consider changes in temperature, salinity and the mixed layer depth. Gradients in both potential temperature and salinity across the shelf break of Prydz Bay present a clear seasonal cycle, first increasing in January, peaking in April and lastly decreasing in June (Fig. 4e, f). The high (low) gradients in austral summer (winter) make the warm events conspicuous (inconspicuous). In other words, the mixed layer is shallow across the austral summer but deepens in the austral winter, consistent with a previous study[52].

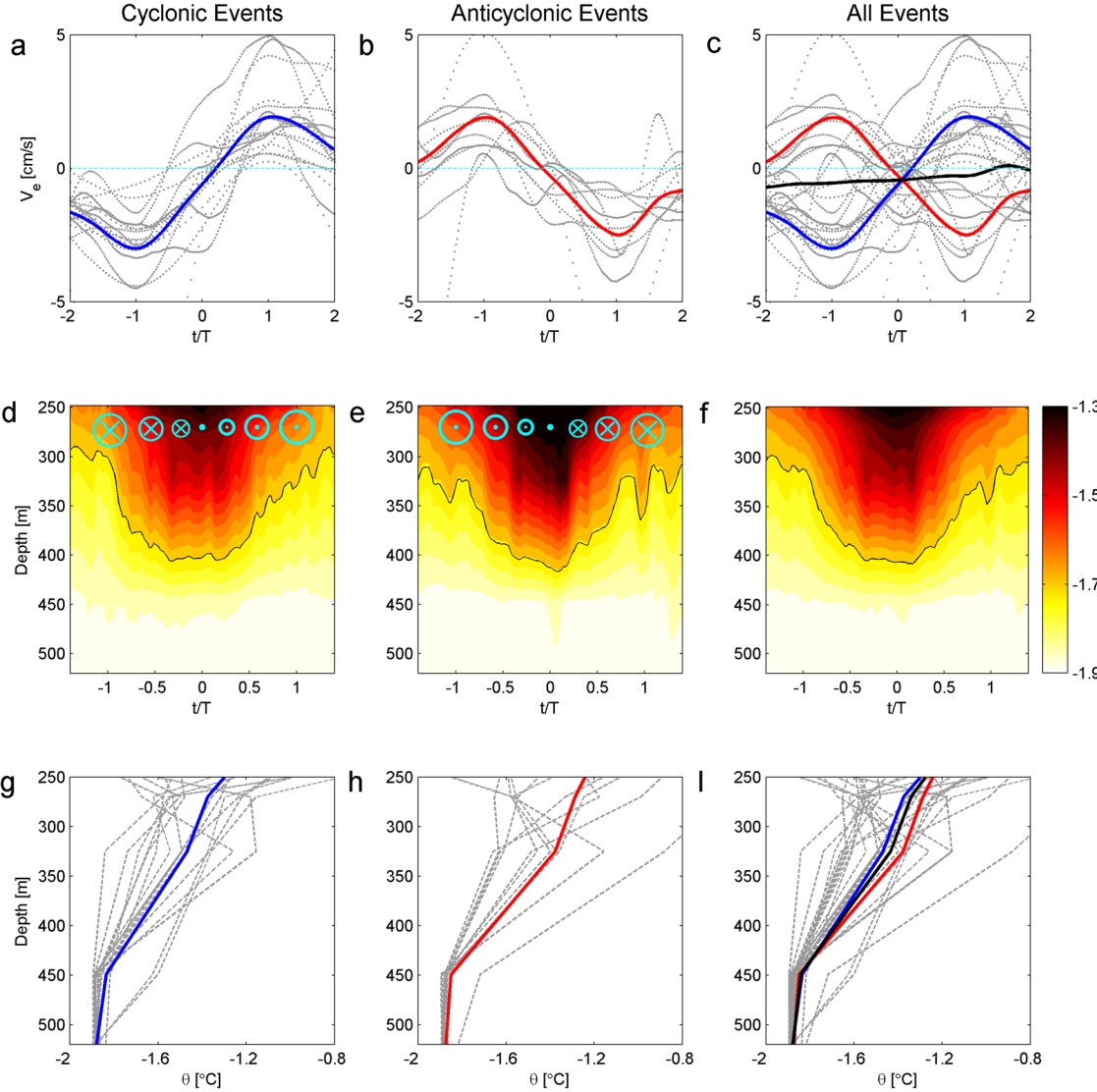

**Fig. 3 | Evolution and structure of cross-shelf eddies.** Composites of cross-stream velocity records ($v_e$) at 270 m (upper panel), eddy processes (middle panel) and vertical structures (lower panel) in potential temperature. (**a**, **d** and **g**) are composites of the cyclonic events (17 events). (**b**, **e** and **h**) are composites of anticyclonic events (11 events). (**c**, **f** and **i**) are composites of all events. Cross-stream velocity records ($v_e$) at 270 m are shown on (**d** and **e**). Black lines are the mean values of blue and red lines in (**c** and **i**). Black lines in (**d**, **e** and **f**) are −1.7 °C contours. The time (x-axis in a-f) is normalized using Eq. (2).

Temperature in the deepening mixed layer remains uniform near the freezing point when there is active sea ice formation driving oceanic convection. The maximum winter mixed layer depth occurs after it is cut-off from the atmosphere by total sea ice cover, with the temperature remaining uniform thereafter. Therefore, no warm events appear in the temperature records during austral winter, despite the continuing presence of eddy intrusions onto the continental shelf. An earlier study[24] suggested a similar process resulted in ongoing, but highly modified CDW intrusions occurring through winter in the Adélie Depression. This supports our argument that the eddies are likely to be occurring through winter, but without a strong temperature signal to identify them.

**Physical processes of warm-eddy formation and transport**

A typical warm intrusion event was observed during the hydrological survey in February 2013 (Fig. 1b, d). The evolution of SSH, geostrophic flow, and wind forcing anomalies describe the process of eddy formation and intrusion onto the continental shelf (Fig. 5a–f and Supplementary Fig. 4). A previous study described the eddy formation mechanism in response to large negative vorticity imparted by the presence of a storm in terms of classical Ekman layer theory[53]. In the upper ocean, strong wind stress curl drives radial outward Ekman transport, resulting in an SSH decrease. Because of the continuity of mass, there must be a strong upwelling underneath the center of the storm to supply this mass flux. Assuming the frictional force is negligible, the flow is primarily cyclogeostrophic below the surface layer, and thus a cyclonic mesoscale eddy formed in the subsurface layer. There are also other examples in the literature that show that wind stress curl can be a mechanism for eddy generation[54,55]. Similarly, two strong cyclones passed north of Prydz Bay from west to east during 26-30 January 2013 (Supplementary Fig. 4), and caused cyclonic eddy formation near the continental slope. The cyclonic winds drove the surface water divergence by Ekman transport, resulting in a continuous decrease in SSH (~3 cm) and an increase of geostrophic flow (15 cm/s) around the eddy in the following week. The intensified cyclonic eddy upwelled the warm mCDW and moved southward into the Prydz channel. The westward ASF near the continental shelf break bent southward influenced by the topographic troughs, and subsequently the eddy deformed and traversed the ASF barrier to transfer relatively warm waters to the shelf (Fig. 5a–f).

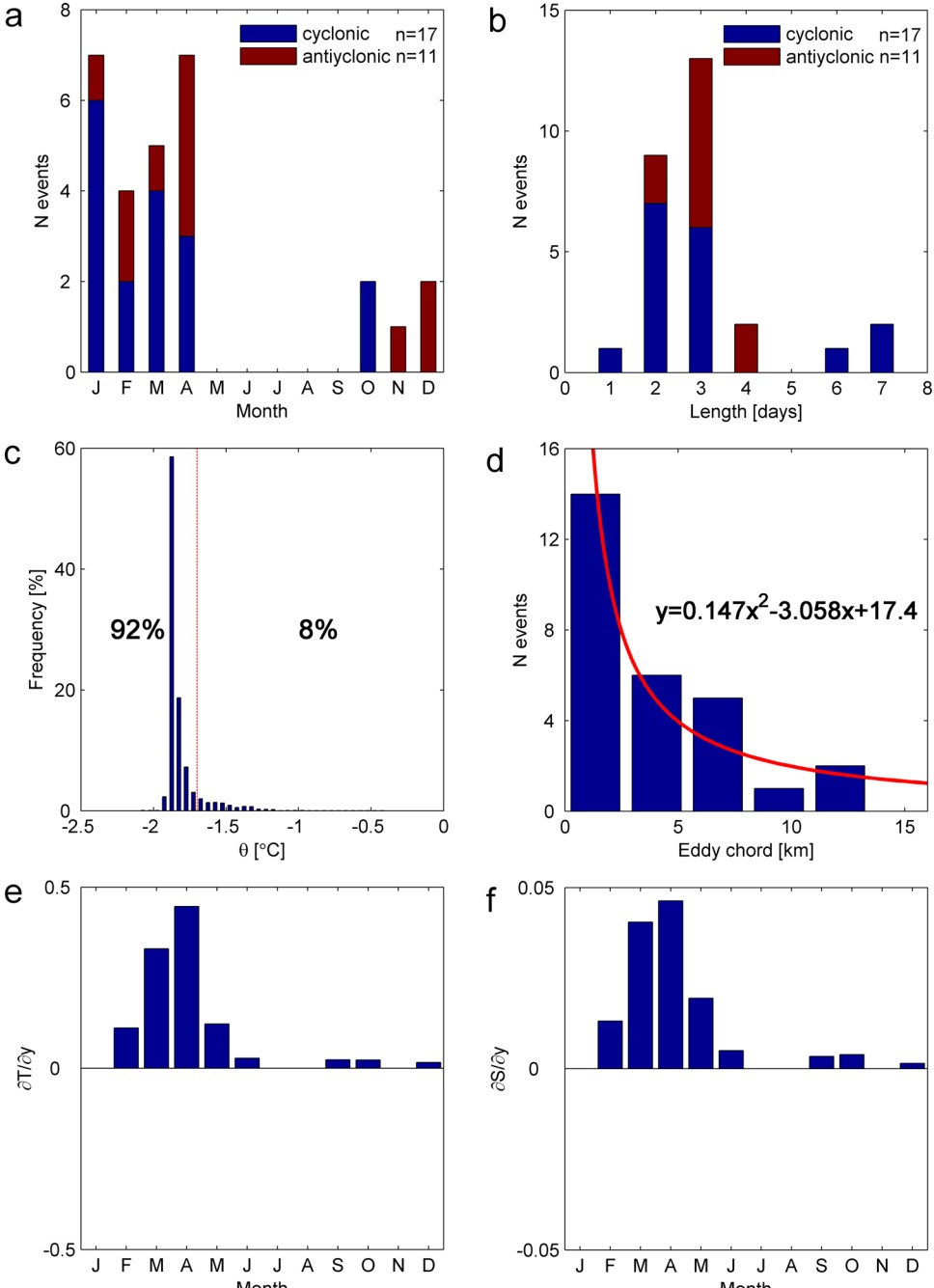

**Fig. 4 | Eddy event characteristics. a** Monthly distribution. **b** Duration-time passing the mooring. Blue and red colors in **a** and **b** are for the cyclonic and anticyclonic events respectively. **c**, Occurrence frequency of potential temperature of mooring records (red dashed line shows −1.7 °C). **d** Horizontal scale of eddy events (fitting curve shown in red line). **e**, **f** are the 200-400 m averaged gradients in potential temperature (unit: °C/degree latitude) and salinity (unit: psu/degree latitude) calculated from elephant seal Conductivity-Temperature-Depth (CTD) data along Prydz Channel (blue boxes in Fig. 1a).

The regional mean changes of SSH and wind forcing help explain the process of the eddy formation and intrusions (Fig. 5g). The consistent cyclonic winds during 26-30 January caused surface divergence and a decrease in SSH that triggered a cyclonic eddy formation. The persistent positive westerly wind from 31 January (~ 5 m/s) increased the offshore Ekman transport and associated upward Ekman pumping near the continental slope, which drove the eddy onto the continental shelf and overcame the ASF barrier. Changes along the 73°E transect between a typical warm intrusion event (mid-February 2013) and the 'normal' case (mid-February 2015) help to further understand the warm-eddy intrusion process (Fig. 5h). Positive temperature anomalies (maxima-1.5 °C) appear in the 100-500 m layer on the continental shelf

between 66.6°S-67°S, while negative temperature anomalies (minima ~ −1.3 °C) are distributed in the 150–700 m depth range on the continental slope between 65.5°S–66.6°S. This suggests that the negative wind stress curl and enhanced westerly winds offshore (Fig. 5g) raise the density surface and decrease the isopycnal slope near the continental shelf break (66.2°S–67°S), driving both the mCDW and ASF upward and southward onto the continental shelf (Fig. 5h).

Here we investigate the influence of zonal winds as a potential physical mechanism for the warm-eddy intrusions. Almost all of the warm events that carry heat onto the continental shelf are associated with wind stress curl changes and enhanced westerlies (Supplementary Figs. 5 and 6). Strong cyclones passed north of Prydz Bay in the first

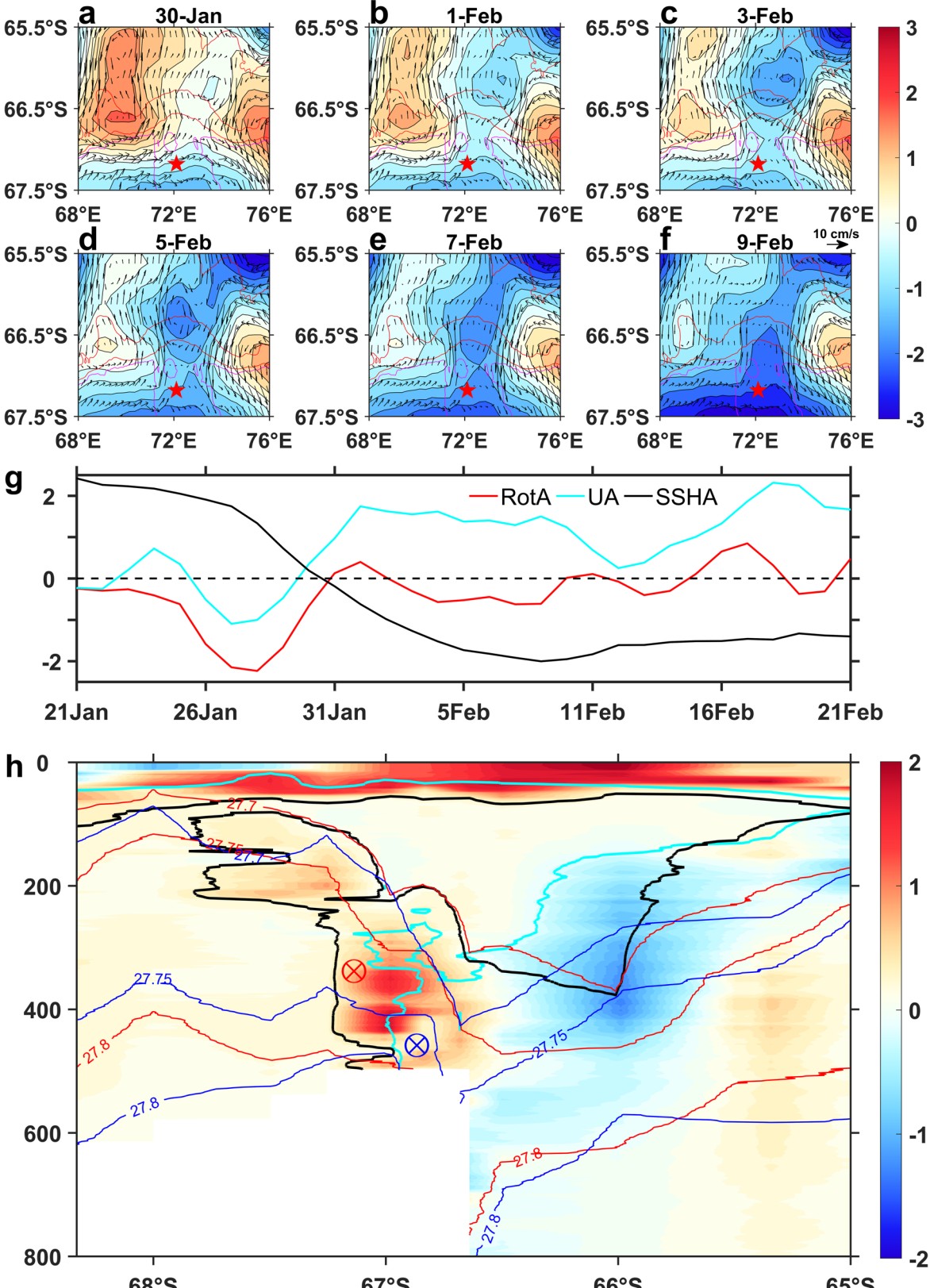

week of the observations (Supplementary Fig. 5b), driving surface water divergence by Ekman transport, resulting in a continuous decrease in SSH (2–5 cm) and an increase in cyclonic eddy formation near the continental slope (Supplementary Fig. 5a). The persistent positive westerly wind (Supplementary Fig. 5c) increased the offshore Ekman transport and associated upward Ekman pumping near the continental slope, which drove the eddies onto the continental shelf and overcame the ASF barrier. The warm anticyclonic eddy events that usually accompany the cyclonic eddies due to the multi-meander of ASF are also driven into the Prydz Bay by the enhanced westerly winds (Supplementary Fig. 6). Regression analyzes between the zonal wind speed, wind stress curl and warm events (warm signals at 270 m) show that the

**Fig. 5 | Eddy formation and intrusion onto the continental shelf. a–f** Evolution of Sea Surface Height (SSH) anomaly (shade and contours, cm) and geostrophic flow anomaly (arrows) during 30-January-9-February in 2013. The mooring location is shown with a red star. The 500 m, 1000 m, 2000m and 3000 m bathymetric contours are shown in red. **g** Evolution of zonal wind anomaly (averaged in the 65ºE-85ºE, 56ºS-66ºS region, cyan line, unit: 3 m·s⁻¹), wind stress curl anomaly (averaged in the 65ºE-85ºE, 56ºS-66ºS region, red line, unit: 2 × 10⁻¹⁰ m·s⁻²) and SSH anomaly (averaged in the 70ºE-74ºE, 65.7ºS-67.1ºS region, black line, unit: cm) during 21-January-21-February in 2013. **h** Potential temperature changes along 73ºE transect between 2013 and 2015 (2013 minus 2015, shade, unit: °C). Black and cyan lines indicate −1.7 °C contours in 2013 and 2015. Red and blue lines are the density surface in 2013 and 2015 respectively. Red and blue circles indicate the southern boundaries Antarctic Slope Front (ASF) (flow from east to west) in 2013 and 2015.

zonal wind (wind stress curl) had a statistically significant (above the 95% confidence level) effect on eddy events with a lead-time of 4-14 days (3-7 days). The maxima of mean regression values over different lead times (4-8 days) for the warm events relative to the zonal wind appear over the 68ºE-78ºE, 60ºS-68ºS region, which covers the transition area between the climatological westerly and easterly wind regimes (Fig. 6a). Most of the warm events occur 4-8 days after the westerly winds strengthened in this transition area (Supplementary Fig. 7a). Negative maximum regression values of warm events relative to wind stress curl appear in the 66ºE–78ºE, 65ºS–68ºS region, just over the continental shelf break of the Prydz channel (Fig. 6b). The cyclonic wind stress curl in this area tends to pump the warm signal upwards onto the Prydz Bay continental shelf (Supplementary Fig. 7b). This analysis suggests that the increasing and poleward shifting westerlies, alongside the resulting cyclonic wind stress curl, are responsible for the warm-eddy intrusions. Additionally, during the 1979-2018 period, both the austral summer (DJF) westerly winds in the 50ºS-67ºS region and the coastal easterly winds south of 67ºS, have strengthened and thus more cyclonic wind stress curl has occurred over the continental shelf break of the Prydz channel (Fig. 6c, d, Supplementary Figs. 8 and 9). Such changes would drive more warm eddies towards the front of the Amery Ice Shelf (north part), where thinning/mass loss is occurring at a significant rate[56].

For Antarctic coastal regions with broad continental shelves, increasing and poleward shifting westerlies increase the offshore Ekman transport and associated upward Ekman pumping near the continental slope, and thus facilitate to shift the mCDW layer upward. Meanwhile, the intensified westerly (easterly) winds offshore (onshore) generate more cyclonic wind stress curl near the continental slope, which also enhances the associated upward Ekman pumping, resulting in an elevated density surface and upward-tilting of the mCDW layer. Topographic troughs in the shelf break region make the ASF bend southward and form loop currents and meanders, with the Ekman pumping contributing further to the cyclonic wind stress curl over the continental shelf break, thus enhancing the onshore warm-eddy intrusions (Figs. 5 and 6). Thereafter, the warm-eddy intrusions join the basin circulation gyres, which supply the ocean heat to the local Antarctic ice shelves. It is worth noting that although most of the eddy intrusion events can be explained by the wind forcing, a number of the eddy events occurred without the trigger of a strong wind stress curl anomaly (Supplementary Fig. 5b). An alternative explanation is that the observed eddies are simply part of the background eddy field, maintained through baroclinic instability, and interact independently with the passing storms.

## Discussion

Previous studies have documented how eddy fluxes compensate for intensified westerlies[41,43–46], the sensitivity of the ACC transport and the overturning circulation as well as relating eddy activity to wind forcing[37–42,45], and hypothesizing the eddy-mediated transport of CDW across the Antarctic Shelf Break[36,37,57]. However, most of these studies are based on models and projections. Detailed observations of how eddies traverse the ASF barrier to facilitate the transfer of relatively warm waters to the shelf regions have been elusive.

Our two-year unique subsurface mooring and hydrographic section observations across 2013-2015 reveal the essential role that eddies play in mCDW intrusions onto the continental shelf of Prydz Bay. Warm events associated with the mCDW intrusions were observed during the austral summer season and were concurrent with cyclonic or anticyclonic eddies. The intrusions are absent in the austral winter due to the deep convection driven by sea-ice formation, despite eddy activity being persistent throughout the year. These persistent warm-eddy intrusions driven by enhanced summer westerly winds bring increased heat flux into Prydz Bay, thereafter contributing to the melting of the Amery ice shelf.

There are several key ice shelf regions surrounding Antarctica that have a broad continental shelf like Prydz Bay, and therefore have a similar environment for the warm-eddy intrusion processes documented by our study. Our results imply that cross-shelf eddy intrusions are one of the dominant processes for water exchange around the Antarctic basins, and suggest they may play a greater role in ice shelf melting than previously thought.

Previous studies have predicted the future changes in the Southern Hemisphere surface westerly wind stress and curl based on observation and simulation[58,59]. Given the prediction that Southern Ocean westerlies will continue to strengthen and shift poleward[58,59], warm-eddy intrusions onto the continental shelves are also likely to increase, with ramifications for ice shelf melting and global sea level rise. Our study highlights the important role of warm eddies in the melting of Antarctica's ice shelves, both now and into the future. The exact processes underlying the formation and transportation of eddies remain a subject of ongoing debate. Further research is essential to clarify the topical concepts presented in this study. Long-term situ observations programs around key Antarctic shelf regions are vital to effectively monitor the variability of ocean heat supply. There is also a critical need to implement eddy-resolution and associated processes into new coupled modeling studies of the Southern Ocean that incorporate Antarctic ice shelves.

## Methods
### Data collections
Hydrological sections data and subsurface mooring observations (Lamont-Doherty Earth Observatory–First Institute Oceanography collaboration) were collected and conducted by the Chinese National Antarctic Research Expedition (CHINARE) from 2013 to 2015. Elephant seal CTD data was accessed from the Marine Mammals Exploring the Oceans Pole to Pole (MEOP) Consortium[60]. All of the associated wind-forcing analyzes were based on daily wind data from the European Centre for Medium-Range Weather Forecasts (ECMWF) Reanalysis-interim (ERA-interim) products, which were able to capture both the high-frequency variability (e.g., changes in storminess) and long-term changes associated with the wind forcing. Two-satellite merged SSH and geostrophic flow data with spatial resolution 0.25°×0.25° was collected from the Archiving, Validation, and Interpretation of Satellite Oceanographic (AVISO).

### Eddy flow estimation
Following previous studies[61,62], all mooring records were de-tided using T_TIDE and subsequently low-pass filtered with a 20-hour period to subtract the inertial frequency variability (Supplementary Fig. 10). Given that the observed horizontal flow $\vec{u}$ can be recognized as the sum of background flow $\vec{U}$ and eddy flow $\vec{u}_e$, the eddy flow then becomes

$$\vec{u}_e = u_e + iv_e = (\vec{u} - \vec{U})e^{-i\phi(t)} \tag{1}$$

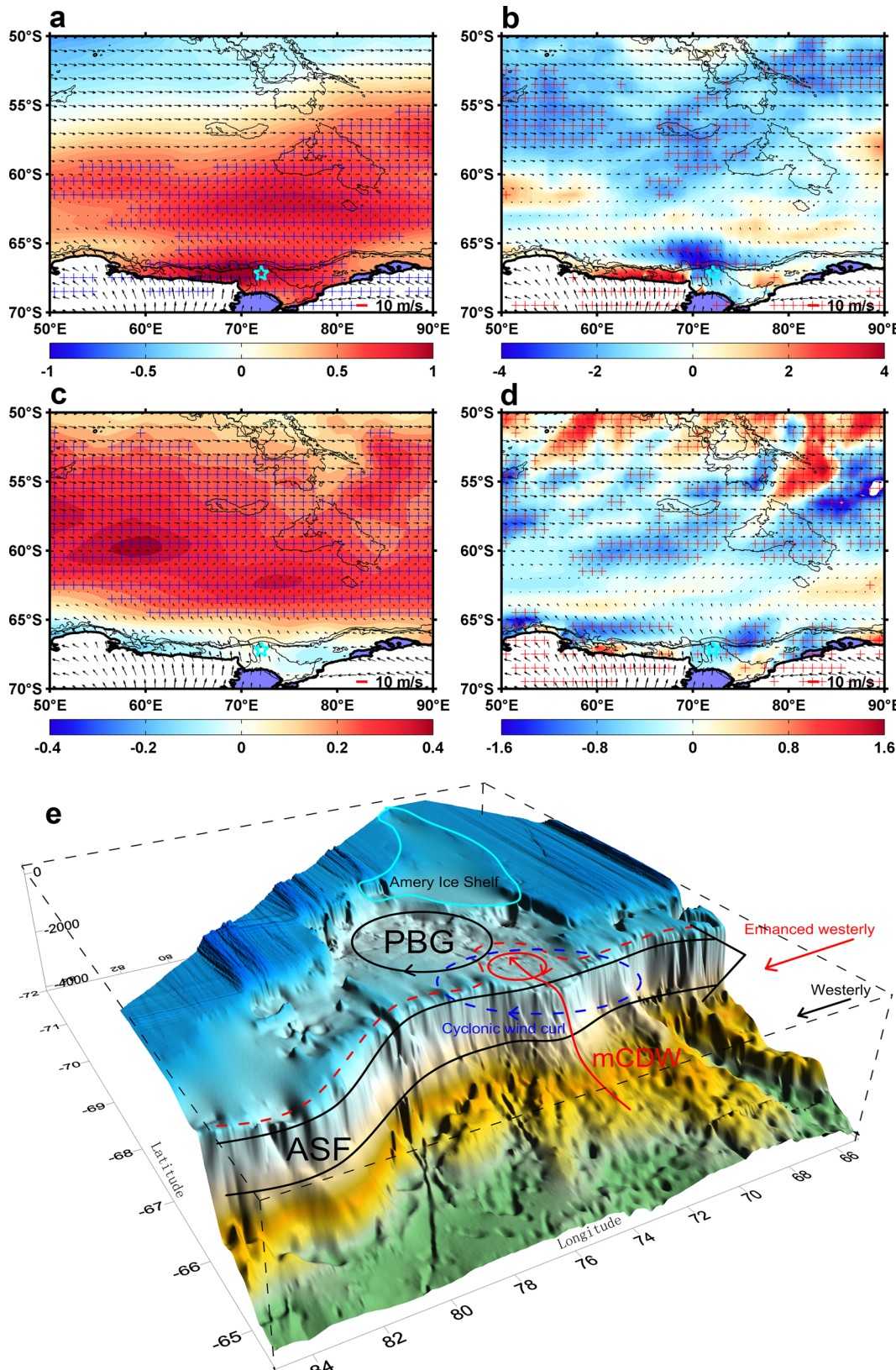

where $\phi$ is the direction of background flow, and $u_e$ and $v_e$ are the along-stream component and cross-stream component of the eddy flow, respectively. The background flow $\vec{U}$ is taken as the 5-day low-pass velocity[7,63].

To compare the eddy signature over different durations, we normalized the time coordinate, where the temperature signal peaks, as

$$t' = (t - t_m)/T_{v\,\mathrm{max}} \tag{2}$$

**Fig. 6 | Physical processes of warm-eddy formation and transport. a** Regression of warm events (potential temperature records at 270 m) on zonal wind (shade, $10^{-2}$ °C /(m·s$^{-1}$), wind leads warm events for 4-8 days). **b** Similar to **a**, but for warm events with wind stress curl (shade, °C /($10^{-10}$m·s$^{-2}$)). **c** Zonal wind trend in austral summer (DJF) during 1979-2018 period (shade, m·s$^{-1}$/(10years)). **d** Similar to **c**, but for wind stress curl (shade, ($10^{-10}$m·s$^{-2}$) /(10years)). Climatology winds are shown with arrows. The 500 m, 1000 m, and 2000m bathymetric contours are shown in black. The mooring location is shown with cyan star. Values over 95% confidence levels are shown with blue or red crosses. **e** Schematic physical processes of warm-eddy formation and transport. Prydz Bay Gyre (PBG) and Antarctic Slope Front (ASF) are shown in black lines. The red dashed line indicates the response of ASF to the enhanced and southward-shifted westerly winds. The Blue dashed line represents cyclonic wind stress curl. Warm eddies and upward-tilting of the modified Circumpolar Deep Water (mCDW) are shown in red solid lines.

where $t$ is the time, $t_m$ is the center time between the two absolute maximum cross-stream velocities in a single event, and $T_{v\mathrm{max}}$ is the time period between those two velocities' maxima.

## Eddy heat flux estimation

The poleward heat flux is calculated based on the background flow and the potential temperature anomalies. Assuming the area of mCDW ($\sim 4.95 \times 10^7$ m$^2$) intrusion into the Prydz Channel observed in 2013 (Fig. 1d) as the mean state. Given the meridional component of background velocity ($1.6 \pm 0.3$ cm/s) during the warm events, the poleward mCDW flux associated with eddy intrusions is around $0.79 \pm 0.15$ Sv.

To estimate the possible heat gain on the Antarctic ice shelf, we assume that all the eddy heat is available for the Amery Ice Shelf melt in Prydz Bay. Following a previous study[16], the annual basal melt rate of the section of the Amery Ice Shelf directly impacted by mCDW inflow due to the eddy heat transport is estimated as

$$m = \frac{\rho F c_p (T_f - T) t}{\rho_i L A_i} \tag{3}$$

where $\rho$ is ocean density (1027 kg m$^{-3}$), $F$ is the mCDW flux into the ice shelf cavity ($\sim 0.79 \pm 0.15$ Sv), $c_p$ is the ocean heat capacity (4000 J kg$^{-1}$ C$^{-1}$), $T_f$ is the pressure freezing temperature ($-2.14$ °C) at 345 dbar (the mean ice shelf draft for the area of the ice shelf base where mCDW can potentially drive basal melt), $T$ is the mean temperature of the mCDW (waters with temperatures over $-1.8$ °C), $t$ is the number of seconds when the temperature above $-1.8$ °C in a year, $\rho_i$ is the density of ice (920 kg m$^{-3}$), $L$ is the latent heat of ice (334,000 J kg$^{-1}$), and $A_i$ is the area of the ice shelf base where mCDW can potentially drive basal melt (12,773 km$^2$). Negative melt rate means the volume of ice is decreasing. Assuming that all the eddy heat associated with this inflow is available for melt, it would result in an average basal melt rate of 3.1 m of ice per year.

## Data availability

Data to support this article are available from the National Centers for Environmental Information (https://www.ncei.noaa.gov/access/metadata/landing-page/bin/iso?id=gov.noaa.nodc:0173518), the ECMWF (https://www.ecmwf.int/en/forecasts/datasets/browse-reanalysis-datasets), and the AVISO (https://www.aviso.altimetry.fr/en/data), the MEOP Consortium (http://www.meop.net).

## Code availability

The code of T_TIDE analysis is available at https://www.eoas.ubc.ca/~rich/t_tide/t_tide_v1.4beta.zip. All codes used to produce the figures are available from the corresponding author on request.

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

## Acknowledgements

This work was supported by the National Key R&D Program of China (2018YFA0605701, 2022YFC2807604), the NSFC grants (41876231), the program of Impact and Response of Antarctic Seas to Climate Change (IRASCC 01-01-01 A), the Basic Scientific Fund for National Public Research Institutes of China (2018S02, GY0222Q03), the NSF grants (ANT-1043669, PLR-1443444), and the Australian Research Council. We are grateful to the staff of Chinese Arctic and Antarctic Administration (CAA) and the crew of the R/V XUELONG and XUELONG2.

## Author contributions

L.G., X.Y., and W.C. conceived the study. L.G. analyzed the data and created the figures. L.G., W.C., and G.D.W. wrote the paper. L.G., X.Y., W.C., G.G., W.Y., J.S., F.Q., Z.W., and G.D.W. contributed to discussing and improving the manuscript.

## Competing interests

The authors declare no competing interests.
