## [Peer Review File · Nature Communications]

Persistent warm-eddy transport to Antarctic ice shelves driven by enhanced summer westerliesEditorial Note: Parts of this Peer Review File have been redacted as indicated to remove third-party material where no permission to publish could be obtained.

REVIEWER COMMENTS

Reviewer #1 (Remarks to the Author):

Review of "Persistent warm-eddy transport to Antarctic ice shelves driven by enhanced summer westerlies", by Libao Gao, Xiaojun Yuan, Wenju Cai, Guijun Guo, Weidong Yu, Jiuxin Shi, and Guy D. Williams.

General Comments

This study primarily uses data from a two year mooring in the Prydz Channel on the Antarctic continental shelf, but also data from hydrological sections, instrumented seals, satellite altimetry, and atmospheric reanalysis, to study the characteristics of episodic intrusions of relatively warm modified Circumpolar Deep Water (mCDW) onto the continental shelf. The authors use the mooring data to identify individual "eddy-like warm events" and estimate the rotation, frequency, and size of these eddy driven impulses of mCDW. The warm events only occur during the summer months and the authors show that that is due to the temperature gradient along the trough (driven by surface mixed layer processes) rather than eddies only occurring during the summer. The authors then present correlations between the off shore westerlies, and the wind stress curl at the shelf break, with the strength of the individual warm events to show that strengthened winds promote these intrusions.

Because of the importance of mCDW intrusions on, not only the melting of the Amery Ice Shelf that is local to this study, but on the melting of Antarctic ice shelves in general, I feel the subject is certainly worth the attention of Nature Communications. The mooring is a fantastic data set to study this issue. While this is not the first high temporal resolution data on eddies advecting across an Antarctic shelf (see comment on lines 286-287 below), there are very few moorings that have captured this data, and much (most?) of the previous results are along the west Antarctic Peninsula, which does not have the dynamic barrier of the Antarctic Slope Front that exists here.

However, I think more could be done to put these results in a broader context in terms of whether the eddies are a significant contributor to the heat budget on the continental shelf. For example, the authors do compute a mean heat flux (lines 126-128) for the warm events. Can the authors estimate a total heat flux due to these warm events across all of Prydz Channel from this? Is this total heat flux comparable to the heat known to be lost through the nearby polynyas? The heat lost to melting of the near ice shelf front portions of the Amery Ice Shelf?

Also, I'm a little confused by the authors discussion of the formation of the off-shelf cyclonic eddy described on lines 189-201. Are the authors proposing that most of the mesoscale eddies that carry heat onto the continental shelf originate off the shelf due to wind divergence or was this a one time event? I could easily be wrong, but while winds can impact the ability (as also shown in this paper) of eddies to get onto the shelf, I was under the impression that most of the mesoscale eddy formation relevant to heat advection onto the Antarctic continental shelf was thought to be due to baroclinic instability (e.g. Stewart and Thompson, GRL, 2015; McKee et al., JPO, 2019). If this is not the case, I think this is an interesting finding that could be emphasized more. Do the authors know of any literature supporting the idea of these particular eddies originating more from wind divergence?

I have several more specific comments and suggestions below, although many of them are very minor editorial comments. I think this paper can eventually be published, but it does require some significant revision first.

Specific Comments

Line 22: I think "contribution" in this context implies that the ice shelf melt water directly increases sea level. If the authors agree, then I suggest changing "contribution to" to "impact on" or something that does not directly imply that ice shelf meltwater raises sea level.

Line 39: Same comment about "contribution" as above.

Line 42: Since the ASF does not go all the way around Antarctica (see the schematic in ref. 18, Thompson et al., 2018), I suggest changing "onto the continental shelf" to "onto most of the continental shelf".

Line 84: Suggest changing "southward at" to "southward to".

Line 93: I think "moorings" should be "mooring".

Line 97: A negative sign is missing before "1 deg C".

Line 106: I don't think "between April 2014 and January 2015" is correct for the highest frequency of events. Several months in that time frame (May-Sep 2014) have no events.

Line 111: Is the 3-8 cm/s amplitude during the entire time series or just the warm events?

Lines 115-116: I think the authors should add a sentence (perhaps not here, but in the methods section or Figure 2 caption) about exactly how the heat fluxes are calculated.

Lines 126-128: I don't understand how the comparison of the lateral heat flux at one specific point onto the continental shelf to the estimated heat gain of SAMW, computed on a much much coarser (2 degrees by 4 degrees I think) scale is relevant. As mentioned above, it would be nice to see a comparison of this to other estimates of heat gain/loss on the Antarctic continental shelf though.

Line 130-131: As mentioned above, I think more needs to be done to show that the "eddy-induced heat transport into Prydz Bay has an important impact on the mass balance of the Amery ice shelf".

Line 137: Suggest adding "cross-stream" before "velocity signal" (if this is referring to the cross-stream velocity).

Line 163-165: What is the Rossby deformation radius from the observations in this area?

Lines 171-174 and Figure 4e and f: What direction is the gradient in? It makes more sense to me if it is in the cross shelf break direction and the denominator of the y-axis label in the figure is "dy". However, the figure caption says "across Prydz Channel". Also (for Figure 4e-f), what are the units for the gradients?

Lines 177-180: How deep do the winter mixed layers get here? Do they drop down into the 200-400 m depth range used for computing the gradients?

Lines 222-227 and Figure 6a (similar question for Figure 6b): Apologies if this should be obvious, but I'm not sure what is meant by the wind leads warm events by 4-8 days with respect to the figure. Is the plotted regression for a specific lead time (e.g. 6 days)? Is it a regression computed over several different lead times from 4 to 8 days. Is the plot showing the regression for the lead time with the maximum correlation at each point with (potentially) a different lead time at each location?

Lines 231-233: I agree that this analysis suggests a relation between increased westerlies and changes in the wind stress curl with warm intrusions. However, have the authors shown a relationship with the poleward shift of the westerlies? I realize the poleward shift will change the curl, but I don't

know if that has been explicitly demonstrated at this point in the manuscript.

Line 233: Suggest adding "the" before "1979-2018 period".

Lines 237-238: Is there a paper or any data showing if the melt rate of the Amery Ice Shelf has been increasing over some portion of the last 40 years?

Line 256: Most of these references are looking at the eddies off the continental shelf. Do any of these actually discuss "eddy-mediated transport of CDW across the Antarctic Shelf Break"? If not, suggest adding '33' to the references here, since that one does explicitly look at eddy transport across the shelf break.

Lines 260-261: Is "seasonally enhanced" the best wording? The authors do show that the summer winds and wind stress curl are increasing in time. However, "seasonally enhanced" implies (to me anyway) that the winds are stronger in the summer than during the rest of the year and that is why the intrusions come in during the summer. There is no discussion of the summer wind strength vs. other seasons (apologies if there is and I missed it) and the authors show (lines 171-182) that the seasonality of the intrusions is driven by the seasonality of the temperature gradient.

Line 283: Suggest changing "role of eddies play" to either "role of eddies" or "role eddies play".

Lines 286-287: These are not the first observations of persistent eddies across an Antarctic shelf break: Moffat et al., 2009 (referenced in this manuscript), Martinson and McKee, 2012 (also referenced here), and Couto et al., 2017 all have nice detailed observations of eddy transport of warm mCDW across an Antarctic shelf. What might be different here is that all those observations were on the WAP, where the ASF does not exist. There are other locations where the ASF exists where authors have observationally studied the impact of eddies transporting mCDW onto the shelf (Nost et al., 2011; Foppert et al., 2019), but I don't think they had the fine temporal resolution of measurements this study does. I think the authors need to be a little more specific than just stating that this is the "first observations of persistent eddies across an Antarctic shelf break".

Line 292: Suggest changing "likely increase" to "likely to increase".

Figure 1 caption, Line 522: Are the authors sure reference 17 (Jacobs, 1991) is the correct one for the polynya definitions?

Extended Data Fig. 3: What are the units for the wind anomaly vectors and the SLP anomaly contours?

Extended Data Fig. 4: Why are there are so many more data points in Figure 4a vs. 4b? Both are using the same set of warm events, correct?

Reviewer #2 (Remarks to the Author):

The manuscript by Dr. Gao and collaborators presents evidence of seasonal warm water intrusions in the Prydz Bay channel based on temperature observations from a 2-year moored record and hydrographic sections collected in the vicinity of the mooring 2 years apart. Con-current velocity observations from the mooring further show warm intrusions to be associated with either cyclonic and anticyclonic eddies. The authors explained the presence/absence of this warm intrusions is linked to two factors: (1) local westerly winds and the associated negative wind stress curl upwelling lifting CDW onto the continental shelf, (2) the seasonal cycle of surface buoyancy loss eroding the warm temperature signal despite the eddies being present year-round.

The authors have done a great job extracting information from the in-situ record and combining their results with the remote sensing analysis. I believe results presented in this paper will be of interest to the Nature Communications readership. But the manuscript still needs a fair bit of work before it can be published. The introduction and discussion lack focus, there are a number of arguments that are unclear or incomplete, and there is missing information throughout the manuscript. Detailed comments are provided below.

Major comments

1. The introduction and parts of the discussion need to be brought into focus. The authors spend a significant amount of text discussing the response of the ACC to changes in westerly winds. While the ACC and the ASC may share some similarities in terms of their eddy overturning circulations, they are distinct features of the Southern Ocean circulation and to my knowledge there is no direct link between increased eddy activity in the ACC and increase eddy shedding of the ASC. If the authors feel there is such link, they need to articulate it much more clearly. Furthermore, the analysis the authors present is based on a relatively short time series, and while it adds to our understanding of the processes that govern cross-shelf exchange, it can't in itself address the question of long-term changes in the system (see also comment 3).

The introduction could also spend more time framing the question about delivering heat to the Antarctic margin in the context of East Antarctica specifically. A lot of the existing literature on the topic of cross-shelf exchange, including modelling and observational evidence of the role eddies play, is based on the West Antarctic Peninsula and the Amundsen Sea. The margins around East Antarctica are significantly different with primarily warm shelves in the west, and weak or missing ASF, and cold/dense in the East, with a well-defined ASF.

Last with regards to the introduction I would encourage the authors to be more selective on the references they used. Throughout the introduction the authors cite long lists of papers on relation to one topic, without explicitly describing what the findings are, and at times including references that are not really appropriate for the topic. For instance, in line 43 the authors say "have captured this process" following a discussion about the role of the ASF as a barrier to cross-shelf transport, while the references that follow described either papers that focus on the circulation and processes within the continental shelf, or in the case of the Amundsen Sea, the role of the undercurrent delivering heat to the shelf. Another clear example is found in lines 60-61. Other references that would be quite relevant to the subject of eddy transport across the shelf-break, such as Stewart et al 2015, are in the reference list but are actually not cited in the text.

2. In my view, there are two processes that are tangled throughout the manuscript. The first is the wind driven uplift of the warm CDW that delivers the water up to the shelf break, and the speculated eddies that can then carry the warm water across the shelf break. Collectively, these two processes deliver heat onto the continental shelf. The correlation analysis provides some evidence about the forcing mechanism behind the first (the uplifting of warm water). However, the manuscript lacks a clear explanation for the second (the formation and evolution of the eddies).

3. Throughout the manuscript the authors appear to link observed trends in the Southern Ocean winds (figures 6 d and e) to warm water intrusions in Prydz bay. But the manuscript provides no evidence of what the link is. The in situ observations capture a seasonal process, and with just 2 years' worth of data it would be quite challenging to detect a trend. And then in line 238 the authors mistakenly claim that the Amery Ice Shelf is thinning at a significant rate and cite the Smith et al 2019!! When in fact Smith and collaborators conclude that the Amery has a relatively small height change, and it's of mass gain not of mass loss. The manuscript would be much more convincing if the authors simply concentrated in the processes they observe, and how they operate at a regional scale, rather than try to connect them with long term changes in the large scale winds.

Minor comments

1. Perhaps more appropriate than "Persistent" in the title would be "Seasonal", as the intrusions are only observed in the summer time.
2. Line 21. The word "prescribed" implies that this is the only factor controlling the delivery of heat, when in reality the circulation within the shelf, buoyancy fluxes, and the geometry of the ice-shelf cavity are just as important. Word-count permitting, I would suggest the authors rephrase this first sentence in the abstract.
3. Line 78 – Mentioning the seal data gives the reader the impression it is another source of data used in this study. Please remove.
4. Line 88-89- Here the intrusion is described as "weaker", when later on with regards to figure 5, it's considered as the "normal" conditions when the intrusion is absent. Please correct for consistency.
5. Line 103, the reader would benefit from a brief description (in relation to the relevant literature cited) of how these events are detected.
6. Lines 108-109. Could the authors clarify how would the method would work in the presence of a northward background flow? Wouldn't a method based on the sequential detection of a pair of opposite sign velocity anomalies depend on the direction of the background flow? Please clarify.
7. Line 119, with regards to heat fluxes the reader would benefit from a brief description of how these fluxes are calculated. Estimating heat transported by and eddy, derived from a single point eulerian measurement is not a trivial. This is due to the fact that in the time mean, the velocity of an eddy advected past a mooring site is zero (that is for a perfectly symmetric eddy). Some other non-trivial aspects of the calculation would be over which length scale to integrate the single point measurement.
8. Lines 127-128. It's unclear what the reference to Subantarctic Mode Water is here for. Consider removing.
9. Line 191. Typo: replace describes by describe.
10. Lines 192-194. Can the authors elaborate on how exactly they observe the "passing" cyclones in Figure 5? I suspect the main signal in the SSH anomaly field has a larger footprint than the small domain being plotted. My guess is that if the authors replotted the same six panels, removing the local mean value, all panels would look much more similar. This large-scale signal is superimposed on the eddy pattern could be the response to the large storm the authors show in the extended data figure 3.
11. Lines 198-200. The authors should consider rescaling panels a-f in figure 5. Due to the small size of the vectors, and the dark shades of blue, it is quite hard to see what the authors describe here as the ASF bending southward and the eddies traversing the ASF.
12. Line 204. Could the authors clarify the mechanism for how surface divergence triggers cyclonic eddies?
13. Lines 222. What does the 3-7 day lag in parenthesis refer to? And why is it different than the 4-14 day?
14. Line 284. Typo. Please remove "the" (in "the Antarctica").
15. At the time of review the link to the mooring/CTD didn't work.
16. Can the authors please supply DOI's for the complete, revised reference list?

Figures

Figure 1

- Could the authors use a different marker type to show in the map which exact stations are plotted in the sections in panels b-e. Having some markers along the top of the sections illustrating where the data actually is would also help pick up features that may have result from interpolation.
- Figure 5, is quite important to the authors arguments. Instead of plotting the small area around the stations in the dashed blue line, they could plot the domain used in Figure 5. It could help providing context to Figure 5.

Figure 5

- Vectors in the southern part of panels a-f are not distinguishable from the background. Could the

authors clarify why are different averaging boxes used for the different variables.

- And in line 558, please clarify what the "southern boundaries of the ASF" and how are they specified.
- Please, add units to panel g.

Figure 6

- Correlations values in figure 6b seem odd. In Panel b these range from [-1,1] why [-4,4] in panel b? And could the authors clarify what exactly is plotted in these two panels? Is it the maximum correlation from a lagged-correlation calculation? The authors briefly mention the lag in the manuscript. If this is the case, it would be useful to provide the corresponding figure with the lags, at least as supplementary material.
- Also, is the correlation analysis carried out only for the summer months? It is not described in the text nor in the figure caption. If it was only calculated using summer months, how did the authors handle the lag? Please clarify. If in the other hand, the correlation included all months, then can the authors explained how can they rule out the role of the seasonal buoyancy loss in the correlation (I would expect seasonal winds would correlate well with seasonal winter heat loss).

Reviewer #3 (Remarks to the Author):

This paper presents the observations of warm eddies with subsurface mooring near Prydz Bay. The eddy-induced heat transport is important to Antarctic ice shelf melt. This is an interesting topic. This manuscript is well organized. However, further process studies are required to define the formation and recurrence of eddies on the mooring site and the characteristics of ASF flow and exchanges.

Comments:

1. The eddies' characteristics are derived from the spiking patterns of temperature time series and spiral velocity records. The salinity is also important to trace the water mass of eddies. The T-S diagram might be useful to show the intrusion of warm water and warm eddies.
2. Line 118-121: The poleward heat flux is calculated from the warm events. From Fig.3c and Extended Data Fig.1c,d, the net cross-stream and meridional component of background velocity are very small. It might be overestimated or underestimated without consideration of the path and movement speed of eddies.
3. Line 133-134 : the warm-eddy intrusion is assumed from the mCDW cross ASF. More detailed numerical modeling will be needed to understand the exchanges between the eddy field and upwelling and cross-shelf transport. (Ref. 53 Liu et al's model results are useful.)
4. Line 188-195: Fig. 5a-f. The mooring is almost located at the center of the saddle shape of SSH. The rotation of the velocity at the mooring site was related to large-scale SSH oscillation. The mesoscale variability near mooring also could be filaments.
5. Line 264-280: "We show that increasing and poleward shifting westerlies, in conjunction with the cyclonic wind stress curl, cause upward-tilting of the mCDW layer that contribute to facilitating warm-eddy intrusions." Both the cyclone and anticyclone eddies were observed. However, the mechanism of formation cyclone and anticyclone eddy is not clarified.

Point-by-point response to Reviewer #1:

Reviewer #1 (Remarks to the Author):

Review of "Persistent warm-eddy transport to Antarctic ice shelves driven by enhanced summer westerlies", by Libao Gao, Xiaojun Yuan, Wenju Cai, Guijun Guo, Weidong Yu, Jiuxin Shi, and Guy D. Williams.

General Comments

1. This study primarily uses data from a two-year mooring in the Prydz Channel on the Antarctic continental shelf, but also data from hydrological sections, instrumented seals, satellite altimetry, and atmospheric reanalysis, to study the characteristics of episodic intrusions of relatively warm modified Circumpolar Deep Water (mCDW) onto the continental shelf. The authors use the mooring data to identify individual "eddy-like warm events" and estimate the rotation, frequency, and size of these eddy driven impulses of mCDW. The warm events only occur during the summer months and the authors show that that is due to the temperature gradient along the trough (driven by surface mixed layer processes) rather than eddies only occurring during the summer. The authors then present correlations between the off shore westerlies, and the wind stress curl at the shelf break, with the strength of the individual warm events to show that strengthened winds promote these intrusions.

Because of the importance of mCDW intrusions on, not only the melting of the Amery Ice Shelf that is local to this study, but on the melting of Antarctic ice shelves in general, I feel the subject is certainly worth the attention of Nature Communications. The mooring is a fantastic data set to study this issue. While this is not the first high temporal resolution data on eddies advecting across an Antarctic shelf (see comment on lines 286-287 below), there are very few moorings that have captured this data, and much (most?) of the previous results are along the west Antarctic Peninsula, which does not have the dynamic barrier of the Antarctic Slope Front that exists here.

Response: Thanks for your comments. A few observational studies have described cross-slope eddies near the Antarctic continental shelves (Nost et al., 2011;

Thompson et al., 2014; Foppert et al., 2019). But precisely how eddies traverse the ASF barrier to facilitate the transfer of relatively warm waters to the shelf regions remains poorly understood, which is why we made it the focus of our study.

(1) Nøst, O. A., Biuw, M., Tverberg, V., Lydersen, C., Hattermann, T., & Zhou, Q., Smedsrud, L. H., and Kovacs, K. M. Eddy overturning of the Antarctic Slope Front controls glacial melting in the Eastern Weddell Sea. *Journal of Geophysical Research: Oceans* (2011). doi:10.1029/2011JC006965.

(2) Thompson, A., Heywood, K., Schmidtko, S. et al. Eddy transport as a key component of the Antarctic overturning circulation. *Nature Geosci.*, 7, 879–884 (2014). <https://doi.org/10.1038/ngeo2289>.

(3) Foppert, A., Rintoul, S. R., & England, M. H. Along-slope variability of cross-slope eddy transport in East Antarctica. *Geophys. Res. Lett.*, 46, 8224–8233. (2019). <https://doi.org/10.1029/2019GL082999>.

2. However, I think more could be done to put these results in a broader context in terms of whether the eddies are a significant contributor to the heat budget on the continental shelf. For example, the authors do compute a mean heat flux (lines 126-128) for the warm events. Can the authors estimate a total heat flux due to these warm events across all of Prydz Channel from this? Is this total heat flux comparable to the heat known to be lost through the nearby polynyas? The heat lost to melting of the near ice shelf front portions of the Amery Ice Shelf?

Response: Agreed. We have compared our estimates with the heat transport of mCDW to the Amery ice shelf (Herraiz-Borreguero et al., 2015). Assuming that all of the eddy heat is available for melting the Amery Ice Shelf in Prydz Bay, the poleward heat flux associated with eddy intrusions (0.79 ± 0.15 Sv) would result in an average basal melt rate of 3.1 ± 0.7 m of ice per year (see Methods). It is roughly 50% more than the previously estimated annual basal melt rate (2.0 ± 0.5 m/year) driven by mCDW flux into the Amery Ice Shelf cavity (Herraiz-Borreguero et al., 2015).

The poleward heat flux is calculated based on the background flow and the potential temperature anomalies. We assume that the area of mCDW ($\sim 4.95 \times 10^5$

m²) intrusion into the Prydz Channel observed in 2013 (Fig.1d) is the mean state. Given the meridional component of background velocity (1.6±0.3 cm/s) during the warm events, the poleward mCDW flux associated with eddy intrusions is around 0.79±0.15 Sv.

To estimate the possible heat gain on the Antarctic ice shelf, we assume that all the eddy heat is available for melting the Amery Ice Shelf in Prydz Bay. Following Herraiz-Borreguero et al. (2015), the annual basal melt rate of the Amery Ice Shelf due to the eddy heat transport is estimated as

$$m = \frac{\rho F c_p (T_f - T) t}{\rho_i L A_i},$$

where ρ is ocean density (1027 kg m⁻³), F is the mCDW flux into the ice shelf cavity (~0.79±0.15 Sv), c_p is the ocean heat capacity (4000 J kg⁻¹ C⁻¹), T_f is the pressure freezing temperature at 345 dbar where mCDW can potentially drive basal melt (-2.14 °C), T is the mean temperature above -1.8 °C of the mCDW, t is the number of seconds when the temperature above -1.8 °C in a year, ρ_i is the density of ice (920 kg m⁻³), L is the latent heat of ice (334,000 J kg⁻¹), and A_i is the area of the ice shelf base where mCDW can potentially drive basal melt (12,773 km²). Assuming that all of the eddy heat associated with this inflow is available for melt, it would result in an average basal melt rate of 3.1 m of ice per year. We have added these in the Methods section in our manuscript. We have revised our manuscript accordingly with changes shown in green color.

Herraiz-Borreguero, L., R. Coleman, I. Allison, S. R. Rintoul, M. Craven, and G. D. Williams (2015), Circulation of modified Circumpolar Deep Water and basal melt beneath the Amery Ice Shelf, East Antarctica, J. Geophys. Res. Oceans, 120, 3098 – 3112, doi:10.1002/2015JC010697.

3. Also, I'm a little confused by the authors discussion of the formation of the off-shelf cyclonic eddy described on lines 189-201. Are the authors proposing that most of the mesoscale eddies that carry heat onto the continental shelf originate

off the shelf due to wind divergence or was this a one time event? I could easily be wrong, but while winds can impact the ability (as also shown in this paper) of eddies to get onto the shelf, I was under the impression that most of the mesoscale eddy formation relevant to heat advection onto the Antarctic continental shelf was thought to be due to baroclinic instability (e.g. Stewart and Thompson, GRL, 2015; McKee et al., JPO, 2019). If this is not the case, I think this is an interesting finding that could be emphasized more. Do the authors know of any literature supporting the idea of these particular eddies originating more from wind divergence?

Response: Thanks for your suggestion. We have conducted the composition analysis of all warm events (Extended Data Fig. 5), which indicates that most of the mesoscale eddies that carry heat onto the continental shelf originate off the shelf due to wind forcing. Strong cyclones (negative wind stress curl anomaly) passed north of Prydz Bay during the first week (Extended Data Fig.5b), driving the surface water divergence by Ekman transport, resulting in a continuous decrease in SSH (2-5 cm) and increase cyclonic eddy formation near the continental slope (Extended Data Fig.5a). The intensified cyclonic eddy upwelled the warm mCDW and moved southward into the Prydz channel. The persistent positive westerly wind (Extended Data Fig.5c) increased the offshore Ekman transport and associated upward Ekman pumping near the continental slope, which drove the eddy onto the continental shelf and overcame the ASF barrier. The westward ASF near the continental shelf break bent southward, influenced by the topographic troughs, and subsequently the eddy deformed and traversed the ASF barrier to transfer relatively warm waters to the shelf. This phase of cyclonic eddy formation and transport onto the continental shelf lasted for almost three weeks (Extended Data Fig.5a, Fig. 5a-f). This analysis suggests that the increasing and poleward shifting westerlies, alongside the resulting cyclonic wind stress curl, are responsible for the warm-eddy intrusions (Extended Data Fig.5a-c).

Moreover, in a manner similar to Fig. 5a-f and Extended Data Fig.4, Figure S1 and Figure S2 show another case of eddy formation and intrusion on to the

continental shelf driven by wind forcing.

The increased upwelling and southward transport driven by wind forcing causes a baroclinic instability, and spawns eddies that counteract the steepening of the isotherms due to wind-driven Ekman pumping in the surface mixed layer (Marshall and Radko, 2003). The process we described is consistent with previous studies (e.g. Stewart and Thompson, GRL, 2015; McKee et al., JPO, 2019). Wang et al. (2016) observed a three-dimensional structure of ocean cooling and eddy formation induced by Pacific tropical cyclones. Furthermore Marshall and Radko (2003) and Stewart and Thompson (2012) also support the concept that the cross-shelf eddies originate more from wind forcing.

- (1) Wang, G., Wu, L., Johnson, N., & Ling, Z. (2016). *Observed three-dimensional structure of ocean cooling induced by Pacific tropical cyclones. Geophysical Research Letters, 43(14), 7632-7638. doi:10.1002/2016gl069605.*
- (2) Marshall, J., and T. Radko (2003), *Residual-mean solutions for the Antarctic Circumpolar Current and its associated overturning circulation, J. Phys. Oceanogr., 33(11), 2341 -2354.*
- (3) Stewart, A. L., & Thompson, A. F. (2012). *Sensitivity of the ocean's deep overturning circulation to easterly antarctic winds. Geophysical Research Letters, 39(18).*

Extended Data Fig. 5 | Composition analysis of all warm events. **a**, Evolution of SSH anomaly (averaged in the 70-74°E, 65.7-67.1°S region, black line, unit: cm). **b**, Evolution of wind stress curl anomaly (averaged in the 65-85°E, 56-66°S region, red line, unit: $2 \times 10^{-10} \text{ m}^2$). **c**, Evolution of zonal wind anomaly (averaged in the 65-85°E, 56-66°S region, cyan line, unit: 3 m s^{-1}). Red lines are the ensemble mean. X-axis is days.

Figure S1. Evolution of SSH anomaly (shade and contours, cm) and geostrophic flow anomaly (arrows) during 20-January~28-January in 2014. The mooring location is shown with red star. The 500m, 1000m, 2000m and 3000m bathymetric contours are shown in red.

Figure S2. Evolution of wind forcing anomaly during 16-January~27-January in 2014. (wind stress curl anomaly, shade, unit: $10^{-10} \text{ m s}^{-2}$; wind anomaly, arrows; Sea Level Pressure anomaly, contours).

I have several more specific comments and suggestions below, although many of them are very minor editorial comments. I think this paper can eventually be published, but it does require some significant revision first.

Specific Comments

- Line 22: I think "contribution" in this context implies that the ice shelf melt water directly increases sea level. If the authors agree, then I suggest changing "contribution to" to "impact on" or something that does not directly imply that ice shelf meltwater raises sea level.

Response: As suggested by Reviewer #1 and Reviewer #2, we have rephrased the

first sentence in the abstract as “The offshore ocean heat supplied to the Antarctic continental shelf by warm eddies has the potential to greatly impact the melting rates of ice shelves and subsequent global sea level rise”.

5. Line 39: Same comment about "contribution" as above.

Response: we have rephrased this sentence.

6. Line 42: Since the ASF does not go all the way around Antarctica (see the schematic in ref. 18, Thompson et al., 2018), I suggest changing "onto the continental shelf" to "onto most of the continental shelf".

Response: agreed.

7. Line 84: Suggest changing "southward at" to "southward to".

Response: we have changed the text accordingly.

8. Line 93: I think "moorings" should be "mooring".

Response: Yes.

9. Line 97: A negative sign is missing before "1 deg C".

Response: Yes.

10. Line 106: I don't think "between April 2014 and January 2015" is correct for the highest frequency of events. Several months in that time frame (May-Sep 2014) have no events.

Response: sorry to cause confusion. We mean that the highest frequency of events occurred in April 2014 and January 2015. The text should be "in April 2014 and January 2015".

11. Line 111: Is the 3-8 cm/s amplitude during the entire time series or just the warm events?

Response: Yes, we have added "during the entire time series" in the revision.

12. Lines 115-116: I think the authors should add a sentence (perhaps not here, but in the methods section or Figure 2 caption) about exactly how the heat fluxes are calculated.

Response: Agreed. We have added sentences about the heat flux calculation in the methods section.

13. Lines 126-128: I don't understand how the comparison of the lateral heat flux at

one specific point onto the continental shelf to the estimated heat gain of SAMW, computed on a much much coarser (2 degrees by 4 degrees I think) scale is relevant. As mentioned above, it would be nice to see a comparison of this to other estimates of heat gain/loss on the Antarctic continental shelf though.

Response: Agreed. As the response to point 2 above, we have compared our estimates with the heat transport of mCDW to the Amery ice shelf (Herraiz-Borreguero et al., 2015). Assuming that all of the eddy heat is available for the melting of Amery Ice Shelf in Prydz Bay, the poleward heat flux associated with eddy intrusions (0.79 ± 0.15 Sv) would result in an average basal melt rate of 3.1 ± 0.7 m of ice per year (see Methods). This is roughly 50% greater than the previous estimate of annual basal melt rate (2.0 ± 0.5 m/year), driven by the mCDW flux into the Amery Ice Shelf cavity (Herraiz-Borreguero et al., 2015).

14. Line 130-131: As mentioned above, I think more needs to be done to show that the "eddy-induced heat transport into Prydz Bay has an important impact on the mass balance of the Amery ice shelf".

Response: Agreed. As mentioned above, we have compared our estimates with the heat transport of mCDW to the Amery ice shelf (Herraiz-Borreguero et al., 2015). Assuming that all of the eddy heat is available for the melting of Amery Ice Shelf in Prydz Bay, the poleward heat flux associated with eddy intrusions (0.79 ± 0.15 Sv) would result in an average basal melt rate of 3.1 ± 0.7 m of ice per year (see Methods). It is roughly 50% greater than the previous estimate of annual basal melt rate (2.0 ± 0.5 m/year), driven by mCDW flux into the Amery Ice Shelf cavity (Herraiz-Borreguero et al., 2015). This suggests that the eddy-induced heat transport into Prydz Bay has an important impact on the mass balance of the Amery ice shelf.

15. Line 137: Suggest adding "cross-stream" before "velocity signal" (if this is referring to the cross-stream velocity).

Response: Agreed. We have changed the text accordingly.

16. Line 163-165: What is the Rossby deformation radius from the observations in this area?

Response: The Rossby deformation radius from the observations in this area is around 3~4km.

17. Lines 171-174 and Figure 4e and f: What direction is the gradient in? It makes more sense to me if it is in the cross shelf break direction and the denominator of the y-axis label in the figure is "dy". However, the figure caption says "across Prydz Channel". Also (for Figure 4e-f), what are the units for the gradients?

Response: Yes, the direction of the gradient is in the across shelf break direction. The units for the gradients (for Figure 4e-f) are °C/latitude (temperature) and psu/latitude (salinity). We have changed the text accordingly.

18. Lines 177-180: How deep do the winter mixed layers get here? Do they drop down into the 200-400 m depth range used for computing the gradients?

Response: The winter mixed layer here is thicker than 400m, so the 200-400m depth range normally used for computing averaged gradients in potential temperature and salinity is too small in austral winter (see Figure 4e-f).

19. Lines 222-227 and Figure 6a (similar question for Figure 6b): Apologies if this should be obvious, but I'm not sure what is meant by the wind leads warm events by 4-8 days with respect to the figure. Is the plotted regression for a specific lead time (e.g. 6 days)? Is it a regression computed over several different lead times from 4 to 8 days. Is the plot showing the regression for the lead time with the maximum correlation at each point with (potentially) a different lead time at each location?

Response: We calculated the regression between the zonal wind speed and the warm events for several different lead times, respectively, and found that the regression shows a statistically significant (above the 95% confidence level) effect on eddy events with a lead-time of 4-8 days in the transition area. Then we use the mean regressions over different lead times (4, 5, 6, 7, 8 days) to show the relationship between the zonal wind speed and warm events (Figure 6a).

20. Lines 231-233: I agree that this analysis suggests a relation between increased westerlies and changes in the wind stress curl with warm intrusions. However, have the authors shown a relationship with the poleward shift of the westerlies? I

realize the poleward shift will change the curl, but I don't know if that has been explicitly demonstrated at this point in the manuscript.

Response: Here we mainly concentrated in the regional processes that increased the westerlies and the changes in the wind stress curl driving warm eddy intrusions, rather than connecting them with long-term changes in the large-scale winds. The zonal wind trend in austral summer (Figure 6c) is only used for discussion: the poleward shift winds during the 1979-2018 periods might have driven more warm-eddy intrusion into the Prydz Bay.

21. Line 233: Suggest adding "the" before "1979-2018 period".

Response: Agreed.

22. Lines 237-238: Is there a paper or any data showing if the melt rate of the Amery Ice Shelf has been increasing over some portion of the last 40 years?

Response: Smith et al. (2020) provided unified estimates of Antarctic ice mass change from 2003 to 2019 using satellite laser altimetry. Their work showed that the northern part of Amery Ice Shelf was losing mass at an increasing rate (~1 m/year) during the 2003-2019 periods. We have now cited the paper in our manuscript.

23. Line 256: Most of these references are looking at the eddies off the continental shelf. Do any of these actually discuss "eddy-mediated transport of CDW across the Antarctic Shelf Break"? If not, suggest adding '33' to the references here, since that one does explicitly look at eddy transport across the shelf break.

Response: Agreed. We have added the reference here.

24. Lines 260-261: Is "seasonally enhanced" the best wording? The authors do show that the summer winds and wind stress curl are increasing in time. However, "seasonally enhanced" implies (to me anyway) that the winds are stronger in the summer than during the rest of the year and that is why the intrusions come in during the summer. There is no discussion of the summer wind strength vs. other seasons (apologies if there is and I missed it) and the authors show (lines 171-182) that the seasonality of the intrusions is driven by the seasonality of the temperature gradient.

Response: Thanks. We have replaced "seasonally enhanced westerly winds" with "enhanced summer westerly winds" to specify the season we focused on.

25. Line 283: Suggest changing "role of eddies play" to either "role of eddies" or "role eddies play".

Response: Agreed. We have changed "role of eddies play" to "role of eddies".

26. Lines 286-287: These are not the first observations of persistent eddies across an Antarctic shelf break: Moffat et al., 2009 (referenced in this manuscript), Martinson and McKee, 2012 (also referenced here), and Couto et al., 2017 all have nice detailed observations of eddy transport of warm mCDW across an Antarctic shelf. What might be different here is that all those observations were on the WAP, where the ASF does not exist. There are other locations where the ASF exists where authors have observationally studied the impact of eddies transporting mCDW onto the shelf (Nost et al., 2011; Foppert et al., 2019), but I don't think they had the fine temporal resolution of measurements this study does. I think the authors need to be a little more specific than just stating that this is the "first observations of persistent eddies across an Antarctic shelf break".

Response: Agreed. We specified the text as "Our two-year unique subsurface mooring and hydrographic section observations across 2013-2015 reveal the essential role that eddies play in mCDW intrusions onto the continental shelf of Prydz Bay".

27. Line 292: Suggest changing "likely increase" to "likely to increase".

Response: Agreed.

28. Figure 1 caption, Line 522: Are the authors sure reference 17 (Jacobs, 1991) is the correct one for the polynya definitions?

Response: Apologies for the mistake. The reference should be 23 (Williams et al., 2016) here.

29. Extended Data Fig. 3: What are the units for the wind anomaly vectors and the SLP anomaly contours?

Response: The units for the wind anomaly vectors and the SLP anomaly contours are (m/s) and (hpa) respectively.

30. Extended Data Fig. 4: Why are there are so many more data points in Figure 4a vs. 4b? Both are using the same set of warm events, correct?

Response: We show that the poleward shift in the westerlies and associated cyclonic wind stress curl promotes these eddy-induced intrusions. Figure 4b shows the data points with cyclonic wind stress curl, without the data points with anticyclonic wind stress curl.

Point-by-point response to Reviewer #2:

Reviewer #2 (Remarks to the Author):

The manuscript by Dr. Gao and collaborators presents evidence of seasonal warm water intrusions in the Prydz Bay channel based on temperature observations from a 2-year moored record and hydrographic sections collected in the vicinity of the mooring 2 years apart. Con-current velocity observations from the mooring further show warm intrusions to be associated with either cyclonic and anticyclonic eddies. The authors explained the presence/absence of this warm intrusions is linked to two factors: (1) local westerly winds and the associated negative wind stress curl upwelling lifting CDW onto the continental shelf, (2) the seasonal cycle of surface buoyancy loss eroding the warm temperature signal despite the eddies being present year-round.

The authors have done a great job extracting information from the in-situ record and combining their results with the remote sensing analysis. I believe results presented in this paper will be of interest to the Nature Communications readership. But the manuscript still needs a fair bit of work before it can be published. The introduction and discussion lack focus, there are a number of arguments that are unclear or incomplete, and there is missing information throughout the manuscript. Detailed comments are provided below.

Major comments

31. The introduction and parts of the discussion need to be brought into focus. The authors spend a significant amount of text discussing the response of the ACC to

changes in westerly winds. While the ACC and the ASC may share some similarities in terms of their eddy overturning circulations, they are distinct features of the Southern Ocean circulation and to my knowledge there is no direct link between increased eddy activity in the ACC and increase eddy shedding of the ASC. If the authors feel there is such link, they need to articulate it much more clearly. Furthermore, the analysis the authors present is based on a relatively short time series, and while it adds to our understanding of the processes that govern cross-shelf exchange, it can't in itself address the question of long-term changes in the system (see also comment 3).

Response: Agreed. We have re-crafted the 'Introduction' to focus on warm water intrusions and increased eddy shedding of the ASC associated with the poleward intensification of the westerlies. It is all about reinforcing the main messages to make it easier for the reader. Now it is much leaner, more focused and streamlined. We have revised our manuscript accordingly with changes shown in green color.

We agree that it is better to concentrate in the processes observed at a regional scale. Indeed, we mainly focus on the eddy formation and transport on to the Prydz Bay continental shelf driven by regional wind forcing based on subsurface mooring and hydrological section data from 2013-2015. Figure 5, Extended Data Fig. 5 and Figures 6a-b show the influence of zonal winds as a potential physical mechanism for the warm-eddy intrusions in Prydz Bay region. Figures 6c-d and a few additional sentences in our manuscript are provided for speculation that the regional changes in westerly winds in past decades would drive more warm eddies towards the front of the Amery Ice Shelf.

32. The introduction could also spend more time framing the question about delivering heat to the Antarctic margin in the context of East Antarctica specifically. A lot of the existing literature on the topic of cross-shelf exchange, including modelling and observational evidence of the role eddies play, is based on the West Antarctic Peninsula and the Amundsen Sea. The margins around East Antarctica are significantly different with primarily warm shelves in the west, and

weak or missing ASF, and cold/dense in the East, with a well-defined ASF.

Last with regards to the introduction I would encourage the authors to be more selective on the references they used. Throughout the introduction the authors cite long lists of papers on relation to one topic, without explicitly describing what the findings are, and at times including references that are not really appropriate for the topic. For instance, in line 43 the authors say “have captured this process” following a discussion about the role of the ASF as a barrier to cross-shelf transport, while the references that follow described either papers that focus on the circulation and processes within the continental shelf, or in the case of the Amundsen Sea, the role of the undercurrent delivering heat to the shelf. Another clear example is found in lines 60-61. Other references that would be quite relevant to the subject of eddy transport across the shelf-break, such as Stewart et al 2015, are in the reference list but are actually not cited in the text.

Response: Thanks for your suggestions. We have re-crafted the ‘Introduction’ to focus on warm water intrusions and increased eddy shedding of the ASC associated with the poleward intensification of the westerlies. As suggested, we have added some text to introduce that the ASF is especially strong and well-defined around the cold and dense East Antarctic margins rather than around the primarily warm shelves in the west. We have described the topics in ‘Introduction’ more explicitly and selected appropriate references for the topics. Stewart et al (2015) has been cited in the manuscript.

33. In my view, there are two processes that are tangled throughout the manuscript. The first is the wind driven uplift of the warm CDW that delivers the water up to the shelf break, and the speculated eddies that can then carry the warm water across the shelf break. Collectively, these two processes deliver heat onto the continental shelf. The correlation analysis provides some evidence about the forcing mechanism behind the first (the uplifting of warm water). However, the manuscript lacks a clear explanation for the second (the formation and evolution of the eddies).

Response: Thanks for your suggestion. As the response to point 3 above, we have done the composition analysis of all warm events (Extended Data Fig. 5), which indicates that most of the mesoscale eddies that carry heat onto the continental shelf originate off the shelf due to wind forcing. Strong cyclones (negative wind stress curl anomaly) passed north of Prydz Bay in the beginning week (Extended Data Fig.5b), driving surface water divergence by Ekman transport, resulting in a continuous decrease in SSH (2-5 cm) and increase cyclonic eddy formation near the continental slope (Extended Data Fig.5a). The intensified cyclonic eddy upwelled the warm mCDW and moved southward into the Prydz channel. The persistent positive westerly wind (Extended Data Fig.5c) increased the offshore Ekman transport and associated upward Ekman pumping near the continental slope, which overcame the ASF barrier and drove the eddy onto the continental shelf. The westward ASF near the continental shelf break bent southward, influenced by the topographic troughs, and subsequently the eddy deformed and traversed the ASF barrier to transfer relatively warm waters to the shelf. Thus the cyclonic eddy formation and transport on to the continental shelf lasted almost three weeks (Extended Data Fig.5a, Fig. 5a-f). This analysis suggests that the increasing and poleward shifting westerlies, alongside the resulting cyclonic wind stress curl, are responsible for the warm-eddy intrusions (Extended Data Fig.5a-c).

34. Throughout the manuscript the authors appear to link observed trends in the Southern Ocean winds (figures 6 d and e) to warm water intrusions in Prydz bay. But the manuscript provides no evidence of what the link is. The in situ observations capture a seasonal process, and with just 2 years' worth of data it would be quite challenging to detect a trend. And then in line 238 the authors mistakenly claim that the Amery Ice Shelf is thinning at a significant rate and cite the Smith et al 2019!! When in fact Smith and collaborators conclude that the Amery has a relatively small height change, and it's of mass gain not of mass loss. The manuscript would be much more convincing if the authors simply concentrated in the processes they observe, and how they operate at a regional

scale, rather than try to connect them with long term changes in the large scale winds.

Response: Thanks for your suggestion. As the response to point 31 above, we agree that it is better to concentrate in the processes observed at a regional scale. Indeed, we mainly focus on the eddy formation and transport on to the Prydz Bay continental shelf driven by regional wind forcing based on subsurface mooring and hydrological section data from 2013-2015. Figure 5, Extended Data Fig. 5 and Figures 6a-b show the influence of zonal winds as a potential physical mechanism for the warm-eddy intrusions in Prydz Bay region. Figures 6c-d and a few additional sentences in our manuscript are provided for speculation that the regional changes in westerly winds in past decades would drive more warm eddies towards the front of the Amery Ice Shelf.

Figure S3 from Smith et al. (2020) shows the Mass loss from Antarctica (2003-2019). The whole Amery has a relatively small height change. However, the north part of Amery Ice Shelf is thinning at a significant rate, which likely consumes most of the heat inflow taken by the Prydz Bay Gyre. We have rephrased the sentence as ‘Such changes would drive more warm eddies towards the front of the Amery Ice Shelf (north part), which is thinning/losing mass at a significant rate (Smith et al., 2020)’.

REDACTED

Figure S3. Mass loss from Antarctica (2003-2019), From Smith et al. (2020)

Minor comments

35. Perhaps more appropriate than “Persistent” in the title would be “Seasonal” , as the intrusions are only observed in the summer time.

Response: We are inclined to emphasize the “Persistent” warm-eddy transport in summer compared with “rare” warm-eddy transport in other seasons. The title shows the “Seasonal intrusion” by the words “driven by enhanced summer westerlies” .

36. Line 21. The word “prescribed” implies that this is the only factor controlling the delivery of heat, when in reality the circulation within the shelf, buoyancy fluxes, and the geometry of the ice-shelf cavity are just as important. Word-count permitting, I would suggest the authors rephrase this first sentence in the abstract.

Response: We have rephrased the first sentence in the abstract as “The offshore ocean heat supplied to the Antarctic continental shelf by warm eddies has the potential to greatly impact the melting rates of ice shelves and subsequent global

sea level rise”.

37. Line 78 – Mentioning the seal data gives the reader the impression it is another source of data used in this study. Please remove.

Response: The seal data was indeed used in this study. Please see Fig. 4e and Fig. 4f.

38. Line 88-89- Here the intrusion is described as “weaker” , when later on with regards to figure 5, it’ s considered as the “normal” conditions when the intrusion is absent. Please correct for consistency.

Response: We have changed the text accordingly.

39. Line 103, the reader would benefit from a brief description (in relation to the relevant literature cited) of how these events are detected.

Response: The details about the eddy detection are presented in the “Eddy flow estimation” part of the “Methods” section.

40. Lines 108-109. Could the authors clarify how would the method would work in the presence of a northward background flow? Wouldn’ t a method based on the sequential detection of a pair of opposite sign velocity anomalies depend on the direction of the background flow? Please clarify.

Response:

Eddy composites: We search for eddies with equation (1) in the method part of our manuscript. Given that the observed horizontal flow \vec{u} can be recognized as

the sum of background flow \vec{U} and eddy flow \vec{u}_e , the eddy flow then becomes

$$\vec{u}_e = u_e + iv_e = (\vec{u} - \vec{U})e^{-i\phi(t)} \quad (1)$$

Where ϕ is the direction of background flow, u_e and v_e are the along-stream component and cross-stream component of the eddy flow, respectively. The background flow \vec{U} is taken as the 5-day low-pass velocity (Moffat et al., 2009; Martinson and Mckee, 2012).

Please note that u_e and v_e are the along-stream component (along the background velocity) and cross-stream component (cross the background velocity) of the eddy flow, respectively. The background velocity has been accounted for when we calculated the eddy flow. The coordinate axis changes with the background flow. The sketch below shows the difference. All the u_e and v_e are in their own coordinate axis (X-Y). When framed this way, we can get the eddy velocity no matter which direction it is propagating from. See figure (c) and (d), the eddy velocity is relative to the rotation but without any relationship to the background flow. This quantitative analysis can be found in previous studies (Moffat et al., 2009; Martinson and Mckee, 2012).

(1) Martinson, D., McKee, D., 2012. Transport of warm Upper Circumpolar Deep Water onto the western Antarctic Peninsula continental shelf. *Ocean Sci.* 8, 433–442.

(2) Moffat C, Owens B, Beardsley R C. On the characteristics of Circumpolar Deep Water intrusions to the west Antarctic Peninsula Continental Shelf [J]. *Journal of Geophysical Research Oceans*, 2009, 114(C5).

The coordinate axis changes with the background flow (examples)

41. Line 119, with regards to heat fluxes the reader would benefit from a brief

description of how these fluxes are calculated. Estimating heat transported by and eddy, derived from a single point eulerian measurement is not a trivial. This is due to the fact that in the time mean, the velocity of an eddy advected past a mooring site is zero (that is for a perfectly symmetric eddy). Some other non-trivial aspects of the calculation would be over which length scale to integrate the single point measurement.

Response: The poleward heat flux is calculated based on the background flow (v component) and the potential temperature anomalies (v component). The background flow is taken as the 5-day low-pass velocity. We have added a brief description of the heat fluxes calculation in the methods section.

42. Lines 127-128. It's unclear what the reference to Subantarctic Mode Water is here for. Consider removing.

Response: Agreed. As the response to point 2 above, we have compared our estimates with the heat transport of mCDW to the Amery ice shelf (Herraiz-Borreguero et al., 2015). Assuming that all of the eddy heat is available for melting the Amery Ice Shelf, the poleward heat flux associated with eddy intrusions (0.79 ± 0.15 Sv) would result in an average basal melt rate of 3.1 ± 0.7 m of ice per year (see Methods). This is roughly 50% greater than the previous estimate of annual basal melt rate (2.0 ± 0.5 m/year) driven by the mCDW flux into the Amery Ice Shelf cavity (Herraiz-Borreguero et al., 2015).

43. Line 191. Typo: replace describes by describe.

Response: We have changed the text accordingly.

44. Lines 192-194. Can the authors elaborate on how exactly they observe the "passing" cyclones in Figure 5? I suspect the main signal in the SSH anomaly field has a larger footprint than the small domain being plotted. My guess is that if the authors replotted the same six panels, removing the local mean value, all panels would look much more similar. This large-scale signal is superimposed on the eddy pattern could be the response to the large storm the authors show in the extended data figure 3.

Response: Thanks for your suggestion. Figure S4 shows the summer mean SSH distributions as the large-scale pattern. It is clear to see that the regional large-scale circulation is dominated by the westward Antarctic Slope Current (ASC) and the Prydz Bay Gyre (PBG). The strong westward ASC along the continental slope ($\sim 66.6^\circ\text{E}$) is very clear and stable. Summer mean SSH shows strong meridional gradient along the continental slope where the strong westward ASC exists.

The large-scale summer mean SSH pattern can show us the large-scale circulation. However, only the evolution of the SSH anomaly can show us the eddy formation and intrusion process onto the continental shelf (Fig. 5a-f). The local mean value (with high range) will dampen the eddy signals (relatively small compared to the high range mean value). We agree that the large-scale signal could be in response to the large storm shown in the extended data figure 4. But the eddy formation is just the subsequent response to the large-scale signal changes. We focus on the warm eddy formation and intrusion process, so it is better to use the SSH anomaly evolution here.

As shown in Figure 5a-f, the SSH anomaly evolution shows a typical eddy intrusion process onto the continental shelf. The intensified cyclonic eddy upwelled the warm mCDW and moved southward into the Prydz channel. The westward ASF near the continental shelf break bent southward, influenced by the topographic troughs, and subsequently the eddy deformed and traversed the ASF barrier to transfer relatively warm waters to the shelf.

Figure S4. Summer mean SSH distributions

45. Lines 198-200. The authors should consider rescaling panels a-f in figure 5. Due to the small size of the vectors, and the dark shades of blue, it is quite hard to see what the authors describe here as the ASF bending southward and the eddies traversing the ASF.

Response: Thanks for your suggestion. We have tried our best to make Figure 5 as clear as possible. The submission system possibly reduced the resolution of figures automatically when generating the combined PDF file that was available for review. The original figures are clear to see.

46. Line 204. Could the authors clarify the mechanism for how surface divergence triggers cyclonic eddies?

Response: The consistent cyclonic winds caused rapid surface divergence and a decrease in SSH. When the cyclonic winds disappear, the surrounding waters tend to flow back towards center to allow the SSH to recover/re-flatten. Due to the Coriolis force, the flow rotates to the left and thus forms a cyclonic eddy. Wang et al. (2016) observed a three-dimensional structure of ocean cooling and eddy

formation induced by Pacific tropical cyclones.

Wang, G., Wu, L., Johnson, N., & Ling, Z. (2016). *Observed three-dimensional structure of ocean cooling induced by Pacific tropical cyclones. Geophysical Research Letters*, 43(14), 7632-7638. doi:10.1002/2016gl069605.

47. Lines 222. What does the 3-7 day lag in parenthesis refer to? And why is it different than the 4-14 day?

Response: Regression between the zonal wind speed and warm events shows there is a statistically significant (above the 95% confidence level) impact on eddy events with a lead-time of 4-14 days. However, regression between the wind stress curl and warm events shows significant effect on eddy events with a shorter lead-time of 3-7 days.

48. Line 284. Typo. Please remove "the" (in "the Antarctica").

Response: We have changed the text accordingly.

49. At the time of review the link to the mooring/CTD didn't work.

Response: We have updated the data link in the 'Data availability' section.

50. Can the authors please supply DOI's for the complete, revised reference list?

Response: The reference list has been prepared in the format required by Nature Communications journal.

51. Figure 1- Could the authors use a different marker type to show in the map which exact stations are plotted in the sections in panels b-e. Having some markers along the top of the sections illustrating where the data actually is would also help pick up features that may have result from interpolation.

Response: As suggested, we have added markers along the top of the sections in Figure 1b-e illustrating where the data actually is.

52. - Figure 5, is quite important to the authors arguments. Instead of plotting the small area around the stations in the dashed blue line, they could plot the domain used in Figure 5. It could help providing context to Figure 5.

Response: Thanks for your suggestion. The domain used in Figure 5a-f is similar to that used in Figure 5h. This helps to show the eddy evolution and intrusion

process more clearly.

53. Figure 5 - Vectors in the southern part of panels a-f are not distinguishable from the background. Could the authors clarify why are different averaging boxes used for the different variables.

Response: The SSH averaging box is the eddy formation and transport region where it is the most representative, thus we can see the variation of the SSH anomaly clearly. The warm events are relative to the zonal wind anomaly in a large domain (see Figure 6a). Therefore, the averaging boxes are different.

54. - And in line 558, please clarify what the "southern boundaries of the ASF" and how are they specified.

Response: Here we use the southern boundaries of the Antarctic Slope Front (ASF) to show the ASF poleward shift between 2013 and 2015 in Figure 5h. We specified the southern boundaries of the ASF where the meridional gradient in potential density is equal $0.1 \text{ kg/m}^3/\text{latitude}$ and the water temperature is over $-1.7 \text{ }^\circ\text{C}$, which represents the ASF boundaries well. Then we use the mean depth of the southern boundaries to describe the changes of the ASF between 2013 and 2015.

55. - Please, add units to panel g.

Response: Thanks for your suggestion. The zonal wind anomaly, wind stress curl anomaly and SSH anomaly use the same y-axis label in panel g. Therefore, we list the units in the figure captions.

56. Figure 6- Correlations values in figure 6b seem odd. In Panel b these range from $[-1,1]$ why $[-4,4]$ in panel b? And could the authors clarify what exactly is plotted in these two panels? Is it the maximum correlation from a lagged-correlation calculation? The authors briefly mention the lag in the manuscript. If this is the case, it would be useful to provide the corresponding figure with the lags, at least as supplementary material.

Response: Figure 6a shows the regression coefficients of warm events on zonal wind, and Figure 6b shows the regression coefficients of warm events with wind stress curl. Those are the maximum regression coefficients from a

lagged-correlation calculation (wind leads warm events for 4-14 days). As suggested, the square-mean correlation between lead levels in zonal wind and warm events is shown in Figure S5.

REDACTED

Figure S5. Square-mean correlation between lead levels in zonal wind and warm events (95% confidence level are shown in blue line)

57. - Also, is the correlation analysis carried out only for the summer months? It is not described in the text nor in the figure caption. If it was only calculated using summer months, how did the authors handle the lag? Please clarify. If in the other hand, the correlation included all months, then can the authors explained how can they rule out the role of the seasonal buoyancy loss in the correlation (I would expect seasonal winds would correlate well with seasonal winter heat loss).

Response: The warm events mainly occurred in the summer, so we calculated the regression for the summer months. We calculated the regression for each summer respectively and then used the mean state to analyze the warm event processes.

Point-by-point response to Reviewer #3:

Reviewer #3 (Remarks to the Author):

This paper presents the observations of warm eddies with subsurface mooring near Prydz Bay. The eddy-induced heat transport is important to Antarctic ice shelf melt.

This is an interesting topic. This manuscript is well organized. However, further process studies are required to define the formation and recurrence of eddies on the mooring site and the characteristics of ASF flow and exchanges.

Comments:

58. The eddies' characteristics are derived from the spiking patterns of temperature time series and spiral velocity records. The salinity is also important to trace the water mass of eddies. The T-S diagram might be useful to show the intrusion of warm water and warm eddies.

Response: Thanks for the suggestion. We showed the hourly potential temperature at different depth levels (248m, 270m, 325m, 449m, and 520m) in figure 2a in our manuscript. However, we only had one salinity sensor at 520m (SBE37). Figure S6 below shows the T-S diagram of the 520m CTD sensor during 2013-2015. Although the 'warm event' signal is small at the 520m level (near bottom), the T-S diagram still shows the intrusions of warm water clearly. Relatively warm modified Circumpolar Deep Water (MCDW) (temperature $> -1.87\text{ }^{\circ}\text{C}$, 34.49-34.57 psu) intrusions occurred during the austral summer (November-April). During the austral winter (May-October), Winter Water (WW) (temperature $< -1.9\text{ }^{\circ}\text{C}$, 34.50-34.56 psu) reached the bottom layer due to the deep convection across the sea-ice growth season. Meanwhile, there were Shelf Water (SW) (temperature $\sim -1.89\text{ }^{\circ}\text{C}$, salinity $> 34.58\text{ psu}$) present in October. We have now added this T-S diagram as one of the Extended Data Figures and updated the text in the manuscript.

Figure S6. T-S diagram of the 520m observations on the mooring during 2013-2015. Colors represent calendar month. Black lines are the potential density surface.

59. Line 118-121: The poleward heat flux is calculated from the warm events. From Fig.3c and Extended Data Fig.1c,d, the net cross-stream and meridional component of background velocity are very small. It might be overestimated or underestimated without consideration of the path and movement speed of eddies.

Response: We apologize for this typographical error. The units of heat flux we calculated and the estimated heat gain from reference (Gao et al., 2018) should be w/m^2 (not kw/m^2).

60. Line 133-134 : the warm-eddy intrusion is assumed from the mCDW cross ASF. More detailed numerical modeling will be needed to understand the exchanges between the eddy field and upwelling and cross-shelf transport. (Ref. 53 Liu et al' s model results are useful.)

Response: Based on an idealized eddy-resolving coupled ocean-ice shelf model, Zhang et al. (2022) did a great job in investigating the characteristics of mesoscale eddies generated over the continental slope in East Antarctica. In order to focus on mesoscale eddies generated by the ASC and exclude the sea surface influences, e.g., eddies generated by sea ice leads, the sea ice model and the atmospheric forcing are not included. Therefore, their idealized eddy-resolving coupled ocean-ice shelf model maybe not suitable to revisit eddy intrusion driven by wind forcing.

Even high-resolution eddy-resolving model analysis data cannot reproduce eddy intrusion driven by wind forcing well. The simulated data cannot capture the upwelled warm eddy on to the continental shelf (Figure S7 and Figure S8), nor the subsequent evolution of the SSH anomaly (Figure S9 compares with Figure 5a-f) well. It is very challenging for numerical modeling to simulate detailed processes around the Antarctic coastal region, due to the commonality deviation in simulations (e.g., sea ice, ice shelf changes, ice-sea-air interaction, and heat transport). There is a critical need to implement eddy-resolution and associated processes into new-coupled modeling studies of the Southern Ocean.

Zhang L, Liu C, Sun W, Wang Z, Liang X, Li X and Cheng C (2022) Modeling Mesoscale Eddies Generated Over the Continental Slope, East Antarctica. Front. Earth Sci. 10:916398. doi: 10.3389/feart.2022.916398.

Figure S7. **(a and b)**, In-situ potential temperature (shade) and salinity (white contours) from CTD transects in 2013 and 2015 along 73°E transect. **(c and d)** are similar to **(a and b)**, but for eddy- and tide-resolving model HYCOM + NCODA Global 1/12° Analysis data (GLBu0.08, <https://www.hycom.org/dataserver/gofs-3pt0/analysis>).

Figure S8. **(a and b)**, In-situ potential temperature (shade) and salinity (white contours) from CTD zonal transects in 2013 and 2015 along 67.25°S transect. **(c and d)** are similar to **(a and b)**, but for eddy- and tide-resolving model HYCOM + NCODA Global 1/12° Analysis data.

Figure 5a-f. Evolution of SSH anomaly (shade and contours, cm) and geostrophic flow anomaly (arrows) during 30-January~9-February in 2013. The mooring location is shown with red star. The 500m, 1000m, 2000m and 3000m bathymetric contours are shown in red.

Figure S9. Similar to Figure 5a-f, but for the eddy- and tide-resolving model HYCOM + NCODA Global 1/12° Analysis data.

61. Line 188-195: Fig. 5a-f. The mooring is almost located at the center of the saddle shape of SSH. The rotation of the velocity at the mooring site was related to large-scale SSH oscillation. The mesoscale variability near mooring also could be filaments.

Response: Figure S10 shows the summer mean current and temperature distributions at the sea surface, 100m, 200m and 300m levels. It is clear to see that the regional large-scale circulation is dominated by the westward Antarctic Slope Current (ASC) and the Prydz Bay Gyre (PBG). The strong westward ASC along the continental slope (~66.6°E) is very clear and stable at all depths shown. Summer mean SSH shows a strong meridional gradient along the continental slope where the strong westward ASC exists (Figure S4), but without the saddle shape of SSH mentioned by the reviewer. The saddle shaped SSH anomaly only exists during the warm-eddy events. As shown in Figure 5a-f, the SSH anomaly evolution shows a typical eddy intrusion process onto the continental shelf. The intensified cyclonic eddy upwelled the warm mCDW and moved southward into the Prydz channel. The westward ASF near the continental shelf break bent southward, influenced by the topographic troughs, and subsequently the eddy deformed and traversed the ASF barrier to transfer relatively warm waters to the shelf (Fig. 5a-f).

Similar to SSH, there are strong meridional gradients in temperature below

100m along the continental slope (Figure S10, 200m and 300m figures). The ASF bends southward in the Prydz Channel and subsequent eddy formation can take the warm water directly into Prydz Bay. However, there are no obvious zonal gradients in temperature below 100m near the mooring (Figure S10, 200m and 300m figures), so that the large-scale east-west oscillation in SSH near the mooring cannot take warm water from the continental slope onto the continental shelf.

Furthermore, in response to point 2 above, we have compared our estimates with the heat transport of mCDW to the Amery ice shelf (Herraiz-Borreguero et al., 2015). The poleward heat flux associated with eddy intrusions (0.79 ± 0.15 Sv) would result in an average basal melt rate of 3.1 ± 0.7 m of ice per year (see Methods). This is ~50% greater than the annual basal melt rate (2.0 ± 0.5 m/year), driven by mCDW flux into the Amery Ice Shelf cavity, estimated by previous study (Herraiz-Borreguero et al., 2015). This suggests that the eddy-induced heat transport into Prydz Bay has an important impact on the mass balance of the Amery ice shelf.

Figure S10. Summer mean current (arrows) and temperature (shade) distributions at sea surface, 100m, 200m, 300m levels from the eddy- and tide-resolving model HYCOM + NCODA Global 1/12 ° Analysis data. The mooring location is shown with red a star.

62. Line 264-280: “We show that increasing and poleward shifting westerlies, in conjunction with the cyclonic wind stress curl, cause upward-tilting of the mCDW layer that contribute to facilitating warm-eddy intrusions.” Both the cyclone and anticyclone eddies were observed. However, the mechanism of formation cyclone and anticyclone eddy is not clarified.

Response: Thanks for your suggestion. We have done the composition analysis of all warm events with anticyclonic eddies (Extended Data Fig. 6). Similar to cyclonic eddies (Extended Data Fig. 5), warm events with anticyclonic eddies that carry heat onto the continental shelf are also associated with wind forcing. The warm anticyclonic eddy events usually accompany cyclonic eddies (Extended Data Fig. 6a). Strong cyclones (negative wind stress curl anomaly) passed north of

Prydz Bay at the beginning of the observations (Extended Data Fig.6b), driving surface water divergence by Ekman transport and resulting in a continuous decrease in SSH and increase in cyclonic eddies formation near the continental slope. The intensified cyclonic eddy upwelled the warm mCDW and moved southward into the Prydz channel.

The westward ASF near the continental shelf break bent southward, influenced by the topographic troughs. It is worth noting that while the eddy scale is usually smaller than 10km (figure 4d), Prydz Bay channel is wider than 100km. So there can be more than one meander when the ASF turns southward. Sometimes both cyclonic and anticyclonic eddies form on different sides of the meander (Figure S11). The persistent positive westerly wind (Extended Data Fig.6c) increased the offshore Ekman transport and associated upward Ekman pumping near the continental slope, which drove the eddies onto the continental shelf and overcame the ASF barrier. This analysis suggests that the increasing and poleward shifting westerlies, together with the resulting cyclonic wind stress curl, are responsible for the warm-eddy (both cyclonic and anticyclonic eddies) intrusions (Extended Data Fig.5 and Extended Data Fig.6).

Extended Data Fig. 6 | Composition analysis of warm events with anticyclonic eddies. **a**, Evolution of SSH anomaly (averaged in the 70-74°E, 65.7-67.1°S region, black line, unit: cm). **b**, Evolution of wind stress curl anomaly (averaged in the 65-85°E, 56-66°S region, red line, unit: $2 \times 10^{-10} \text{ m s}^2$). **c**, Evolution of zonal wind anomaly (averaged in the 65-85°E, 56-66°S region, cyan line, unit: 3 m s^{-1}). Red lines are the ensemble mean. X-axis is days.

Figure S11. In-situ temperature distribution at 200m observed during the summer cruise in 2013.

REVIEWER COMMENTS

Reviewer #1 (Remarks to the Author):

Review of revised manuscript "Persistent warm-eddy transport to Antarctic ice shelves driven by enhanced summer westerlies", by Libao Gao, Xiaojun Yuan, Wenju Cai, Guijun Guo, Weidong Yu, Jiuxin Shi, Fangli Qiao, Zexun Wei, and Guy D. Williams.

General Comments

I thank the authors for their efforts in response to my earlier comments, including putting the heat flux from the eddies in a broader context with the new estimate of the Amery ice shelf melting due to heat transport by the eddies. However, I do have some questions about this estimate:

1) Are the authors sure the listed area of mCDW (line 343) is correct? This looks too small compared to the extent of the black line in Figure 1d and if the area actually is $4.95e5 \text{ m}^2$, then the poleward mCDW flux (assuming a velocity of 1.6 cm/s) is actually $\ll 0.79 \text{ Sv}$. Could the exponent be a typo and the area actually is $4.95e7 \text{ m}^2$? That looks closer to the extent in Figure 1d and would give a flux of 0.79 Sv.

2) If the melt rate calculated is the extra melt due to extra heat from the eddies beyond an ambient temperature of -1.8 , then I do not understand the $(T_f - T)$ term, where T "is the mean temperature above -1.8C of the mCDW", in equation 3. Shouldn't that term just be $-T$ (i.e. the excess temperature above -1.8)? As written (and with this definition of T), I think this equation would be significantly off.

3) If I'm wrong about point 2 above and T_f really does need to be in the calculation, then I think more needs to be explained why a pressure freezing temperature at 345 dbar was chosen (as was done in Herraiz-Borrequeiro et al. where this equation was introduced).

4) As in Herraiz-Borrequeiro et al., the area used for the melt rate is not the entire Amery ice shelf, but rather is the portion of the ice shelf where H-B et al. thought mCDW can potentially drive basal melt. That is briefly mentioned here in the Methods section (lines 357-358), but should be mentioned in the main text (lines 140-145) where the melt from the eddies is presented and compared to the H-B et al. estimate.

I have several other new questions/comments, but the vast majority of these are very minor. I still think some minor revision is necessary before publication, but I believe all these comments can be easily addressed by the authors.

Specific Comments

Line 49: Do the authors mean that there are significant differences between various regions in East Antarctica or that the East Antarctic regions "differ significantly" compared to the West Antarctic regions?

Line 50: Should "warm" be defined?

Line 51: Suggest adding "waters on the" before "shelves are cold and dense".

Lines 64-66: Are all these references (36-41) appropriate for this sentence? While these studies do show mesoscale eddies transferring CDW poleward, most of these are not eddy resolving over the Antarctic continental shelf and I think Stewart and Thompson (apologies if I'm wrong) is the only one

that could show eddies "overcoming the barrier imposed by the ASF" and transferring "mCDW towards the Antarctic ice shelves".

Line 107: "enable" should be "enables".

Lines 140-143: See comment above about what portion of the ice shelf this calculation is for. Perhaps just add something like "over the portion of the ice shelf thought to be subject to melting from mCDW" right after "ice per year".

Line 144: Need to mention here as well that the 2.0 m/year melt estimate is only over a portion of the Amery Ice Shelf (same area as for the authors' calculation).

Line 179: Can these chord sizes be used to come up with an estimate of the full size of the eddies and then compare that to what is seen in the altimetry?

Line 182: The authors answered my previous question about the Rossby deformation radius in this area in their Response letter (Item 16), but I didn't see that the value (3-4 km) was added to the manuscript anywhere.

Line 219: Suggest changing "help understand" to "help explain".

Lines 236-247 and Extended Data Figures 5 and 6: I just wanted to thank the authors for these additions: I really like the new figures. Have the authors thought about doing one like Extended Data Figure 6 for the warm events with cyclonic eddies?

Lines 263-264: I think the authors can change "is likely to have occurred" to "has occurred". Don't Figure 6d and Extended Data Figures 8 and 9 show that it did actually happen?

Line 300: "infer" should be "imply".

Lines 348-350: Suggest a slight rewrite of the first part of sentence to something like:

Following a previous study^(^54), the annual basal melt rate of the portion of the Amery Ice Shelf directly impacted by mCDW inflow due to the eddy heat transport is..."

Figure 1 caption (lines 591-592): Suggest moving the last sentence about how the polynyas are shown to just after "cover the Prydz Channel" (line 587).

Figure 3 caption (line 607): Suggest adding "(x-axis in a-f)" just after "The time".

Figure 4 caption (line 614): Should the denominator for the units for temperature and salinity be "deg latitude" instead of just "latitude"?

Figure 4 caption (line 615): Should "across" be "along" here? I don't know if the authors agree, but I tend to think of this channel being more in the north-south direction and thus a latitudinal displacement being "along" channel.

Figure 6: I think the information the authors provided in their Response letter (Item 19) to my previous question about how the regressions in 6a and 6b are plotted should be added somewhere (either in the Figure caption or the text).

Reviewer #2 (Remarks to the Author):

First of all, I would like to thank the authors for their very detailed response to my earlier comments. I appreciate they have put a significant amount of effort in addressing the issues I raised. In general I'm quite happy with their responses, and by all means, I would like to see their manuscript published. However, there are still a number of lingering issues, I think need to be addressed before the paper can be accepted. These are aspects of the paper that in my opinion remain highly speculative, and the authors would need to provide further evidence supporting them if they were to remain in the manuscript.

1. First is the eddy formation mechanism proposed by the authors. Mesoscales eddies being formed in response to large negative vorticity imparted by the passing of a storm is a new concept, and it needs to be backed by more evidence. The reference the authors provide in the response to my previous comments (Wang et al., 2016), does not describe the eddy formation mechanism they propose, it simply illustrates the effects of a tropical cyclone in the temperature distribution within the mixed layer (a quick search of the word "eddy" in that paper yielded 0 matches, and all the observational data in the paper is restricted to the top 50m of the water column at most). The eddies the authors are describing in this paper don't appear to have much of a surface signature, instead they show as anomalies in the subsurface properties. There is ample literature on the subject of tropical cyclones interacting with mesoscale eddies, but the formation of the latter is always attributed to baroclinic instability. And note, that the mechanism they proposed is also inconsistent with Extended Data figure 5, where they show SSH continues to decrease in the region (panel a), for approximately 3 weeks after the storm has passed (panel b).

2. The second aspect I remained unconvinced is the fact that the eddy-like features the authors show in the snapshots of SSH anomalies have length scales that are one order of magnitude larger ($O(100\text{km})$) than the length scales the authors estimate from the mooring record ($O(10\text{km})$). To me this scale difference suggests the oscillations captured by the mooring record are different in nature to those observed offshore, and likely the result of different processes. The connection between these two types of eddies relies, according to the authors, relies on the offshore eddies being advected by the Ekman transport. Which is something that again, I don't feel they provide strong evidence or a sound dynamical reasoning for.

3. And lastly, with regards to the discussion on the long-term changes in the wind field, the authors provide convincing evidence of how storms passing over the region immediately offshore of the Prydz Bay channel can uplift isopycnals and drive intrusions of warm water across the shelf break. I can't however see how the predicted strengthening and shift in the westerlies alone could result in enhanced CDW intrusions, unless these long-term changes in the westerlies are also accompanied by either stronger or more frequent/persistent storms in the region. Note that the features capture in panel b in Extended figure 5 have typical duration of ~ 5 days. While the data in Extended Data Fig 9 is given as annual means, and hence unable to capture a possible increase in "storminess".

Reviewer #3 (Remarks to the Author):

The authors have taken care of my previous concerns. The manuscript has improved significantly. I recommend accepting it for publication.

We have revised our manuscript accordingly with changes to the revised manuscript shown in *green italics*.

Point-by-point response to Reviewer #1:

Reviewer #1 (Remarks to the Author):

Review of revised manuscript "Persistent warm-eddy transport to Antarctic ice shelves driven by enhanced summer westerlies", by Libao Gao, Xiaojun Yuan, Wenju Cai, Guijun Guo, Weidong Yu, Jiuxin Shi, Fangli Qiao, Zexun Wei, and Guy D. Williams.

General Comments

I thank the authors for their efforts in response to my earlier comments, including putting the heat flux from the eddies in a broader context with the new estimate of the Amery ice shelf melting due to heat transport by the eddies. However, I do have some questions about this estimate:

1. Are the authors sure the listed area of mCDW (line 343) is correct? This looks too small compared to the extent of the black line in Figure 1d and if the area actually is $4.95e5 \text{ m}^2$, then the poleward mCDW flux (assuming a velocity of 1.6 cm/s) is actually $\ll 0.79 \text{ Sv}$. Could the exponent be a typo and the area actually is $4.95e7 \text{ m}^2$? That looks closer to the extent in Figure 1d and would give a flux of 0.79 Sv.

Response: Yes, the area of mCDW intrusion into the Prydz Channel should be $\sim 4.95 \times 10^7 \text{ m}^2$. We apologize for this typographical error.

2. If the melt rate calculated is the extra melt due to extra heat from the eddies beyond an ambient temperature of -1.8, then I do not understand the $(T_f - T)$ term, where T "is the mean temperature above -1.8C of the mCDW", in equation 3. Shouldn't that term just be -T (i.e. the excess temperature above -1.8)? As written (and with this definition of T), I think this equation would be significantly off.

Response: To clarify, we have changed the text to "*T is the mean temperature of the mCDW (waters with temperatures over -1.8 °C)*". In another words, we defined the waters with temperatures over -1.8 °C to be the mCDW intrusions by eddies. And we assume that all of those intrusions can be transported into the

Amery Ice Shelf cavity for basal melt (i.e. mCDW interacts with the base of the Amery Ice Shelf). Thus, the $T_f - T$ term is the temperature difference between the pressure freezing temperature at 345 dbar and the temperature of the mCDW inflow, which supplies extra heat for basal melt. Thereafter equation 3 is used to calculate the annual basal melt rate of the portion of the Amery Ice Shelf directly impacted by mCDW inflow due to the eddy heat transport. The same equation was used in Herraiz-Borrequeiro et al. (2015).

3. If I'm wrong about point 2 above and T_f really does need to be in the calculation, then I think more needs to be explained why a pressure freezing temperature at 345 dbar was chosen (as was done in Herraiz-Borrequeiro et al. where this equation was introduced).

Response: Agreed. Following Herraiz-Borrequeiro et al. (2015), we have explained why a pressure freezing temperature at 345 dbar was chosen. The text has been updated as " *T_f is the pressure freezing temperature (-2.14 °C) at 345 dbar (the mean ice shelf draft for the area of the ice shelf base where mCDW can potentially drive basal melt)*".

4. As in Herraiz-Borrequeiro et al., the area used for the melt rate is not the entire Amery ice shelf, but rather is the portion of the ice shelf where H-B et al. thought mCDW can potentially drive basal melt. That is briefly mentioned here in the Methods section (lines 357-358), but should be mentioned in the main text (lines 140-145) where the melt from the eddies is presented and compared to the H-B et al. estimate.

Response: Agreed. The main text has been updated as "*Assuming that all of the eddy heat is available for basal melting of the area beneath the Amery Ice Shelf basal melt that the mCDW can potentially access, the poleward heat flux associated with the eddy intrusions (0.79 ± 0.15 Sv) would result in an average basal melt rate of 3.1 ± 0.7 m of ice per year over this section of the ice shelf (see Methods)*".

I have several other new questions/comments, but the vast majority of these are very minor. I still think some minor revision is necessary before publication, but I believe

all these comments can be easily addressed by the authors.

Specific Comments

5. Line 49: Do the authors mean that there are significant differences between various regions in East Antarctica or that the East Antarctic regions "differ significantly" compared to the West Antarctic regions?

Response: We mean that the East Antarctic regions differ significantly compared to the West Antarctic regions. To clarify, we have changed the sentence to *'It is important to note that the characteristics of the continental shelf regions surrounding East Antarctica differ significantly from the West Antarctic regions'*.

6. Line 50: Should "warm" be defined?

Response: Agreed. Following Thompson et al. (2018), the warm shelves are defined where the seafloor of the continental shelf is occupied by waters with temperatures exceeding 0 °C. We have changed the text as *'In the western areas, there are primarily warm shelves (bathed in waters with temperatures exceeding 0 °C) ...'* Thompson et al. (2018) is in our references list.

7. Line 51: Suggest adding "waters on the" before "shelves are cold and dense".

Response: We have changed the text accordingly.

8. Lines 64-66: Are all these references (36-41) appropriate for this sentence? While these studies do show mesoscale eddies transferring CDW poleward, most of these are not eddy resolving over the Antarctic continental shelf and I think Stewart and Thompson (apologies if I'm wrong) is the only one that could show eddies "overcoming the barrier imposed by the ASF" and transferring "mCDW towards the Antarctic ice shelves".

Response: Yes, Stewart and Thompson (2015) is the only reference that refers to eddies "overcoming the barrier imposed by the ASF" and transferring "mCDW towards the Antarctic ice shelves". Other references cited here show mesoscale eddies transferring CDW poleward in the southern open ocean, but do not mention eddies overcoming the barrier imposed by the ASF and transferring mCDW towards the Antarctic ice shelves. We only keep the Stewart and Thompson (2015) paper cited here as suggested.

9. Line 107: "enable" should be "enables".

Response: Agreed.

10. Lines 140-143: See comment above about what portion of the ice shelf this calculation is for. Perhaps just add something like "over the portion of the ice shelf thought to be subject to melting from mCDW" right after "ice per year".

Response: We have changed the text accordingly.

11. Line 144: Need to mention here as well that the 2.0 m/year melt estimate is only over a portion of the Amery Ice Shelf (same area as for the authors' calculation).

Response: Agreed. We have indicated that both calculations are for the same area of the Amery Ice Shelf in the revised manuscript.

12. Line 179: Can these chord sizes be used to come up with an estimate of the full size of the eddies and then compare that to what is seen in the altimetry?

Response: It is difficult to estimate the full size of eddies based on the chord sizes, because it is hard for us to confirm whether the chords cross through the centers of eddies or not (assuming eddies are circular in the horizontal plane). The size of an eddy must be greater than or equal to its chords, but it is not related to its chords. For a given chord, the eddy sizes can be very different (see Figure S1 below). It is better to see the full size of eddies from altimetry data.

Figure S1. Possible eddy sizes for a chord

13. Line 182: The authors answered my previous question about the Rossby deformation radius in this area in their Response letter (Item 16), but I didn't see that the value (3-4 km) was added to the manuscript anywhere.

Response: We have added the values in the new revised manuscript.

14. Line 219: Suggest changing "help understand" to "help explain".

Response: We have changed the text accordingly.

15. Lines 236-247 and Extended Data Figures 5 and 6: I just wanted to thank the authors for these additions: I really like the new figures. Have the authors thought about doing one like Extended Data Figure 6 for the warm events with cyclonic eddies?

Response: Great idea. We have done this before. The figure for the warm events with cyclonic eddies is quite similar to the figure for all warm events (Extended Data Figures 5), in that most of the warm events are cyclonic eddies. Therefore, it might be enough to keep Extended Data Figures 5 and 6 here.

16. Lines 263-264: I think the authors can change "is likely to have occurred" to "has occurred". Don't Figure 6d and Extended Data Figures 8 and 9 show that it did actually happen?

Response: Agreed. Those figures show that it did occur. We have changed the text accordingly.

17. Line 300: "infer" should be "imply".

Response: Agreed.

18. Lines 348-350: Suggest a slight rewrite of the first part of sentence to something like: Following a previous study^(^54), the annual basal melt rate of the portion of the Amery Ice Shelf directly impacted by mCDW inflow due to the eddy heat transport is..."

Response: Agreed. We have changed the text accordingly.

19. Figure 1 caption (lines 591-592): Suggest moving the last sentence about how the polynyas are shown to just after "cover the Prydz Channel" (line 587).

Response: Agreed. We have changed the text accordingly.

20. Figure 3 caption (line 607): Suggest adding "(x-axis in a-f)" just after "The time".

Response: Agreed. We have changed the text accordingly.

21. Figure 4 caption (line 614): Should the denominator for the units for temperature and salinity be "deg latitude" instead of just "latitude"?

Response: Agreed. We have changed the units to "°C/degree latitude" and "psu/degree latitude".

22. Figure 4 caption (line 615): Should "across" be "along" here? I don't know if the authors agree, but I tend to think of this channel being more in the north-south direction and thus a latitudinal displacement being "along" channel.

Response: Agreed. We have changed the text accordingly.

23. Figure 6: I think the information the authors provided in their Response letter (Item 19) to my previous question about how the regressions in 6a and 6b are plotted should be added somewhere (either in the Figure caption or the text).

Response: Agreed. We have added a sentence in the text to clarify how the regressions are plotted.

Point-by-point response to Reviewer #2:

Reviewer #2 (Remarks to the Author):

First of all, I would like to thank the authors for their very detailed response to my earlier comments. I appreciate they have put a significant amount of effort in addressing the issues I raised. In general I'm quite happy with their responses, and by all means, I would like to see their manuscript published. However, there are still a number of lingering issues, I think need to be addressed before the paper can be accepted. These are aspects of the paper that in my opinion remain highly speculative, and the authors would need to provide further evidence supporting them if they were to remain in the manuscript.

24. First is the eddy formation mechanism proposed by the authors. Mesoscales eddies being formed in response to large negative vorticity imparted by the passing of a storm is a new concept, and it needs to be backed by more evidence. The reference the authors provide in the response to my previous comments (Wang et al., 2016), does not describe the eddy formation mechanism they

propose, it simply illustrates the effects of a tropical cyclone in the temperature distribution within the mixed layer (a quick search of the word “eddy” in that paper yielded 0 matches, and all the observational data in the paper is restricted to the top 50m of the water column at most). The eddies the authors are describing in this paper don’t appear to have much of a surface signature, instead they show as anomalies in the subsurface properties. There is ample literature on the subject of tropical cyclones interacting with mesoscale eddies, but the formation of the latter is always attributed to baroclinic instability. And note, that the mechanism they proposed is also inconsistent with Extended Data figure 5, where they show SSH continues to decrease in the region (panel a), for approximately 3 weeks after the storm has passed (panel b).

Response: Based on classical Ekman layer theory, a simple analytical solution of a mesoscale eddy induced by a stationary hurricane in a homogenous ocean was discussed in a previous study (e.g., Lu and Huang, 2010). They described the eddy formation mechanism in response to large negative vorticity imparted by the passing of a storm. In the upper ocean, strong wind stress curl drives radial outward Ekman transport, resulting in a SSH decrease. Because of the continuity of mass, there must be a strong upwelling underneath the center of the hurricane to supply this mass flux. In the subsurface layer immediately below the surface layer, the frictional force is assumed to be negligible, and the flow is primarily cyclo-geostrophic, thus a cyclonic mesoscale eddy formed in the subsurface layer (see Figure S2 below). We have now cited Lu and Huang (2010) in the revised manuscript.

About Extended Data figure 5, please note that the evolution of the SSH anomaly (panel a) is averaged in the region (70-74°E, 65.7-67.1°S) over the continental shelf break and the Prydz Bay channel where eddies intrude onto the continental shelf. And the evolution of the wind stress curl anomaly (panel b) is averaged in the region (65-85°E, 56-66°S) further north to the open ocean where storms usually passed. Both the eddy formation and southward transport need a few days’ time. The storm-induced eddy forms in the open ocean where the storm

passed through, and it is driven southward to the continental shelf break by the westerly winds. The intensified cyclonic eddy upwells the warm mCDW and it moves southward into the Prydz channel. Then the westward ASF near the continental shelf break bent southward, influenced by the topographic troughs, and subsequently the eddy deformed and traversed the ASF barrier onto the shelf (Figures 5a-f). Figures 5a-f and Extended Data Figure 4 show that two strong cyclones passed eastwards, north of Prydz Bay, between 26-30 January 2013 (5 days). Thereafter the eddy formed, intensified and move southward between 30 January-5 February 2013 (6 days). The eddy then deformed with the ASF, bending into the topographic troughs between 6-9 February 2013 (4 days). The entire process for this eddy formation and transport takes more than two weeks.

Moreover, in a manner similar to Figures 5a-f and Extended Data Figure 4, Figure S3 and Figure S4 show another case of eddy formation and intrusion onto the continental shelf driven by wind forcing, which also took approximately two weeks. In addition, new storms may occur again before the eddy transportation is finished, so the SSH can continue to decrease for much longer time.

Lu Z. & Huang R. The three-dimensional steady circulation in a homogenous ocean induced by a stationary hurricane. J. Phys. Oceanogr. 40(7), 1441-1457 (2010).

REDACTED

Figure S3. Evolution of SSH anomaly (shade and contours, cm) and geostrophic flow anomaly (arrows) during 20-January~28-January in 2014. The mooring location is shown with red star. The 500m, 1000m, 2000m and 3000m bathymetric contours are shown in red.

Figure S4. Evolution of wind forcing anomaly during 16-January~27-January in 2014. (wind stress curl anomaly, shade, unit: $10^{-10} \text{ m}\cdot\text{s}^{-2}$; wind anomaly, arrows; Sea Level Pressure anomaly, contours).

25. The second aspect I remained unconvinced is the fact that the eddy-like features the authors show in the snapshots of SSH anomalies have length scales that are one order of magnitude larger ($O(100\text{km})$) than the length scales the authors estimate from the mooring record ($O(10\text{km})$). To me this scale difference suggests the oscillations captured by the mooring record are different in nature to those observed offshore, and likely the result of different processes. The connection between these two types of eddies relies, according to the authors, relies on the offshore eddies being advected by the Ekman transport. Which is something that again, I don't feel they provide strong evidence or a sound dynamical reasoning for.

Response: Firstly, the length scales estimated from the mooring record are the eddy chords (assuming eddies are circular in the horizontal plane). Somewhat similar to point 12 raised by Reviewer #1, it is hard for us to confirm whether the chords cross through the centers of the eddies or not. Indeed, the overwhelming majority of eddy chords do not. So the length scales estimated from the mooring record must be smaller than the length scales seen from the SSH anomalies.

Secondly, the mooring record used to estimate the eddy chord length is observed by a current meter located at 270m (Figure 2b and Figure 3), so the length scales estimated from the mooring record are the eddy chords at the 270m depth layer. Since the three-dimensional structure of eddies are vertical-taper structures (Figure 3d-f), the length scales estimated at 270m must be much smaller than the length scales at the sea surface seen from the SSH anomalies.

Thirdly, there are multiple troughs in the topography of the Prydz Bay channel (see Figure 1a). The westward ASF near the continental shelf break bends southward, influenced by these troughs. There can be multiple meanders when the ASF turns southward, with both cyclonic and anticyclonic eddies formed on different sides of the meanders. So the length scales of the eddies estimated from the mooring record could be much smaller than the length scales seen from the SSH anomalies offshore.

Fourthly, the mooring is located at 67.18°S, approximately 100km from the continental shelf break. Intruding eddies could move eastward with the Prydz Bay Gyre before the entire eddy passes the mooring location. This scenario may also reduce the length scales of the eddies estimated from the mooring record.

Finally, Figures 5a-h show the typical process of eddy formation and intrusion onto the continental shelf driven by the wind forcing, by examining the SSH anomaly, wind forcing, and hydrological section observations across the same time period. The agreement of these independent observations supports our findings. Further examination of this process will require more situ observations and eddy-resolution coupled modeling studies of the Southern Ocean that incorporate Antarctic ice shelves.

26. And lastly, with regards to the discussion on the long-term changes in the wind field, the authors provide convincing evidence of how storms passing over the region immediately offshore of the Prydz Bay channel can uplift isopycnals and drive intrusions of warm water across the shelf break. I can't however see how the predicted strengthening and shift in the westerlies alone could result in enhanced CDW intrusions, unless these long-term changes in the westerlies are also accompanied by either stronger or more frequent/persistent storms in the region. Note that the features capture in panel b in Extended figure 5 have typical duration of ~5days. While the data in Extended Data Fig 9 is given as annual means, and hence unable to capture a possible increase in "storminess".

Response: To clarify, all of the associated wind forcing plots (Figures 6a-d, Extended Data Figure 4, Extended Data Figure 8 and Extended Data Figure 9) are calculated basing on the daily wind data from ERA-interim products, hence it is able to capture variations in "storminess". We have updated the text as "*All of the associated wind-forcing analyses were based on the daily wind data from the European Centre for Medium-Range Weather Forecasts (ECMWF) Reanalysis-interim (ERA-interim) products*" in the Data section of the revised manuscript.

For example, we firstly calculated the wind stress curl for each day across the 1979-2018 periods, then calculated the austral summer (including all the daily data for each year' DJF season) wind stress curl trend based on those daily wind stress curls data. This provided the distributions of wind stress curl trend during the 1979-2018 periods (Extended Data Figure 8b). The same method is used in all of the long-term trend calculations for zonal wind and wind stress curl in our study (Figures 6c-d, Extended Data Figure 8 and Extended Data Figure 9). To subtract the high frequency variability in the variation of austral summer wind during 1979-2018 periods, we calculated the mean values of zonal wind and wind stress curl for each austral summer (including all the daily data for each year' DJF season), then we calculated the long-term trend for the time series (Extended Data Figure 9). To subtract the high frequency variability in the variation of austral

summer wind across 1979-2018, we calculated the mean values of zonal wind and wind stress curl for each austral summer (including all daily data for each years' DJF season), and then we calculated the long-term trend for the time series (Extended Data Figure 9). In reality, the long-term patterns remain relatively consistent, regardless of whether the high-frequency variability is retained or not.

Figures 6c-d, Extended Data Figure 8 and Extended Data Figure 9 were calculated based on the daily wind data. These figures show that both the austral summer (DJF) westerly winds in the 50°S-67°S region, together with the coastal easterly winds south of 67°S, have strengthened and thus more cyclonic wind stress curl has occurred over the continental shelf break of the Prydz channel. Such changes would drive more warm eddies towards the front of the Amery Ice Shelf (north part), where thinning/mass loss is occurring at a significant rate.

Reviewer #1 also agreed that the long-term changes of westerly wind and wind stress curl did actually happen basing on Figures 6c-d, Extended Data Figure 8 and Extended Data Figure 9 (see point 16 raised by Reviewer #1). Auger et al. (2022) shows the EOF modes 1 spatial pattern of westerly wind and wind stress curl (see Figure S5), which are consistent with our results. The strengthening and shift in the westerlies can increase the meridional shear of zonal wind, resulting in stronger cyclonic wind stress curl near the coastal region (Figures 6c-d, Extended Data Figure 8, and Figure S5). So we suggest that "*Given the prediction that Southern Ocean westerlies will continue to strengthen and shift poleward⁵⁷⁻⁵⁸, warm-eddy intrusions onto the continental shelves are also likely to increase, with ramifications for ice shelf melting and global sea level rise*".

In the end, we agree that it is better to concentrate on the processes observed at a regional scale. Indeed, we mainly focus on the eddy formation and transport onto the Prydz Bay continental shelf driven by regional wind forcing, based on subsurface mooring and hydrological section data from 2013-2015. Figure 5, Extended Data Fig. 5 and Figures 6a-b show the influence of zonal winds as a potential physical mechanism for the warm-eddy intrusions into the Prydz Bay region. Figures 6c-d, Extended Data Figure 8 and a few additional

thought-provoking sentences in our manuscript are provided to put forward the hypothesis that the regional changes in westerly winds in past decades could have driven more warm eddies towards the front of the Amery Ice Shelf. We prefer to keep these sentences in the manuscript with a view to prompting further work to address this topical concept.

Auger, M., Sallée, J.-B., Prandi, P., & Naveira Garabato, A. C. (2022). Subpolar Southern Ocean seasonal variability of the geostrophic circulation from multi-mission satellite altimetry. Journal of Geophysical Research: Oceans, 127, e2021JC018096. <https://doi.org/10.1029/2021JC018096>

REDACTED

Figure S5. EOF modes 1 spatial pattern of Wind Stress Curl (left) and Zonal Wind Stress (right) during 2013-2019. (From Auger et al., 2022)

Point-by-point response to Reviewer #3:

Reviewer #3 (Remarks to the Author):

The authors have taken care of my previous concerns. The manuscript has improved significantly. I recommend accepting it for publication.

Response: Thanks.

REVIEWER COMMENTS

Reviewer #1 (Remarks to the Author):

Review of re-revised manuscript "Persistent warm-eddy transport to Antarctic ice shelves driven by enhanced summer westerlies", by Libao Gao, Xiaojun Yuan, Wenju Cai, Guijun Guo, Weidong Yu, Jiuxin Shi, Fangli Qiao, Zexun Wei, and Guy D. Williams.

General Comments

I again thank the authors for their efforts in response to my previous comments. My only comment about any of their reply is with respect to item 12 about chord sizes. I agree that for a given chord length the eddy sizes can be very different. However, the authors do have a distribution of chord lengths. What if one assumes that there is an isotropic distribution of the eddies down the trough with respect to the mooring position (the authors would know better than I if that is a reasonable assumption) and that the eddies are not widely different in size? If so, wouldn't the maximum chord lengths from the distribution, even if still an under-estimate, be at least representative of the eddy diameters? Lilly and Rhines, who the authors cite (62) in the methods section with respect to computing chord lengths, note that it is a chord being estimated this way, and thus under-represents the diameter. However, they also do still use the chord lengths to characterize the size of their observed eddies. I do agree with the authors on some of their points to Reviewer #2 regarding eddy sizes. However, I think it's worth the effort trying to at least get an estimate from the mooring data on the size of the eddies over the mooring, even if there are caveats, to compare to the satellite estimates.

I have a couple of other very minor editorial comments, but in general I'm quite happy with the changes made. I recommend acceptance after the comments about chord length (and the other trivial comments below) are considered.

Specific Comments

Lines 50-51: I think this sentence should have a reference at the end. The Thompson et al. reference (21) used for the next sentence would fit this one as well.

Line 142: Suggest removing "basal melt" from between "Amery Ice Shelf" and "that the mCDW".

Lines 211-213 and reference 54: I think it would help the reader if the authors added a little more to the text summarizing the results of the Lu and Huang study that are relevant here. Perhaps something similar to the first paragraph of their reply to Reviewer #2 in their new Response Letter?

Lines 251-254 and Extended Data Figures 7 a+b: What are the correlation coefficients for the regressions shown in the figures?

Equation 3: Now that I understand the $T_f - T$ term better, should T_f and T be switched? As written, wouldn't this give a negative melt rate?

Figure 6 caption (also an issue in the captions for Extended Data Figures 4, 5, 6, 8, and 9): The units for wind stress curl are correct in the text for 6b, but there's a typo in the text for 6d (should be s^{-2}), not s^2).

Comment on re-revised manuscript "Persistent warm-eddy transport to Antarctic ice shelves driven by enhanced summer westerlies", with respect to the authors reply and revisions to comments from Reviewer #2.

I am not Reviewer #2, but have been asked by the editor my thoughts on the authors response to those comments.

Reviewer comment about eddy formation mechanism: I have to admit to being more familiar with papers either discussing the impacts of ocean eddies on atmospheric storms (especially tropical cyclones) or where wind stress weakens ocean mesoscale eddies (e.g. Xu et al., 2016 and the many references cited there), than ones where storms induce eddies in the ocean. However, there are examples in the literature where it is shown that wind stress curl can be a mechanism for eddy generation (e.g. Stammer et al. 2001, Wang et al., 2008). So, while I don't totally agree with reviewer #2 that "eddies being formed in response to large negative vorticity imparted by the passing of a storm is a new concept", I do think it's an idea that is still an open question and thus deserves more discussion in the main text than just a simple reference to the Lu and Huang paper. I mentioned in my earlier review (I am reviewer #1) that I thought it would help the reader if the authors added a little more to the text summarizing the results of the Lu and Huang study along the lines of what they put in their reply to Reviewer #2. I think perhaps a couple more references about this idea (see examples above) may also help.

Stammer et al., 2001. The role of variable wind forcing in generating eddy energy in the North Atlantic. *Progress in Oceanography*.

Wang et al., 2008. Winter Eddy Genesis in the Eastern South China Sea due to Orographic Wind Jets. *Journal of Physical Oceanography*.

Xu et al., 2016. Work done by atmospheric winds on mesoscale ocean eddies. *Geophysical Research Letters*.

Reviewer comment about length scale of eddy features: I did touch on something similar in my earlier comment about the chord lengths. I agree with the authors point about how "the length scales estimated from the mooring record must be smaller than the length scales seen from the SSH anomalies", but I don't think it should be a huge difference, and I think (given the assumptions I mentioned about circular eddies and an isotropic distribution) a good estimate can be made (or an argument made about why those assumptions are way off).

I can see how the length scales estimated from a mooring at 270 m could be smaller than those estimated from SSH anomalies, but I don't see how it could be "much smaller" (and the ratio could be estimated by the authors based on Figures 3d-f). I'm not sure I understand the physics involved in the authors third (eddies onshelf are smaller than offshelf eddies because of multiple meanders in the ASF?) and fourth (eddies become smaller while transiting through the Prydz Bay Gyre) replies to this point. I do agree with the authors that Figure 5 is illustrative of the eddy formation and intrusion onto the shelf. I still think more discussion in the manuscript is necessary in response to this point.

Reviewer comment about the relationship between the westerlies and CDW intrusions: I think the authors have quite nicely answered reviewer 2's concerns on this point. My only suggestion is that there are several examples of modeling studies showing increasing intrusion of CDW and heat onto the Antarctic continental shelf with strengthening and poleward shifted westerlies that could be cited (perhaps starting with Spence et al, 2014).

Spence et al., 2014. Rapid subsurface warming and circulation changes of Antarctic coastal waters by

poleward shifting winds. Geophysical Research Letters.

Reviewer #2 (Remarks to the Author):

I will again start by thanking the authors for the efforts. In my last review, I raised three points where I still had significant doubts. With regards to the second and third points, on the subject of the scale of the eddies onshore/offshore and on the subject of future trends in wind stress and curl, I am happy with the response the authors have provided to me. However, only minor modifications have been made to the actual manuscript to clarify these points and they could still lead to confusion. For instance, instead of simply stating that the wind analysis is based on daily data, the authors could be more explicit about why they think their wind stress analysis is able to capture changes in storminess. Similarly, a sentence to reconcile the order of magnitude difference between the length scales observed by the mooring and those inferred from altimetry would be useful.

With regards to the formation mechanism of the offshore eddy I strongly encourage the authors to soften the language and present it as speculation. The alternative explanation being that the observed eddies are part of the background eddy field, maintained through baroclinic instability, and can interact with the passing storms. My reasons for questioning the validity of this argument again are the following. The new reference the authors bring up in response to my comment, Lu and Huang (2010), is quite an interesting highly idealized theoretical study. As the authors of the study admit, it should be only considered as a limit, given that hurricanes are very rarely stationary (slow moving at most) and the ocean is stratified. But even then, if one was to accept the assumption of stationarity and homogenous ocean, the mechanism proposed in the study can only explain the formation of cyclones, not the anticyclones. And there is the additional problem of scales, where the scale of the oceanic eddy is an order of magnitude smaller than the scale of the storm that triggers it, when the theory predicts comparable amplitude.

Lastly advection of a subsurface feature by the Ekman flow, something the authors reiterate in their response, without further work to demonstrate it, is in my view speculation.

We have revised our manuscript accordingly with changes to the revised manuscript shown in *green italics*.

Point-by-point response to Reviewer #1:

Reviewer #1 (Remarks to the Author):

Review of re-revised manuscript "Persistent warm-eddy transport to Antarctic ice shelves driven by enhanced summer westerlies", by Libao Gao, Xiaojun Yuan, Wenju Cai, Guijun Guo, Weidong Yu, Jiuxin Shi, Fangli Qiao, Zexun Wei, and Guy D. Williams.

General Comments

1. I again thank the authors for their efforts in response to my previous comments. My only comment about any of their reply is with respect to item 12 about chord sizes. I agree that for a given chord length the eddy sizes can be very different. However, the authors do have a distribution of chord lengths. What if one assumes that there is an isotropic distribution of the eddies down the trough with respect to the mooring position (the authors would know better than I if that is a reasonable assumption) and that the eddies are not widely different in size? If so, wouldn't the maximum chord lengths from the distribution, even if still an under-estimate, be at least representative of the eddy diameters? Lilly and Rhines, who the authors cite (62) in the methods section with respect to computing chord lengths, note that it is a chord being estimated this way, and thus under-represents the diameter. However, they also do still use the chord lengths to characterize the size of their observed eddies. I do agree with the authors on some of their points to Reviewer #2 regarding eddy sizes. However, I think it's worth the effort trying to at least get an estimate from the mooring data on the size of the eddies over the mooring, even if there are caveats, to compare to the satellite estimates.

Response: Yes, the distribution of chord lengths is an isotropic distribution, and even the maximum chord lengths are still an under-estimate compared to the full size of eddies at the sea surface. As suggested, we have now added an estimation on the size of eddies over the mooring compared to the satellite estimates. We have also discussed the possible reasons for the under-estimate of eddy chords

over the mooring record. The new text added in the manuscript is as follows: *"It is worth noting that the eddy chords estimated from the mooring record are around 10%~50% of the full eddy size observed from the altimetry data. There are three possible explanations for this. Firstly, the mooring record used to estimate the eddy chords is observed by a current meter located at 270m (Figure 2b and Figure 3). Since the three-dimensional shape of eddies are vertical-taper structures (Figure 3d-f), the eddy chords at 270m must be much smaller than the length scales at the sea surface. Secondly, the majority of the eddy chords do not cross through the center of the eddies (assumed to be circular in the horizontal plane). Thirdly, the mooring is located at 67.18°S, approximately 100 km from the continental shelf break. Intruding eddies could move eastward with the Prydz Bay Gyre, preventing the entire eddy from passing over the mooring location. All three reasons explain the reduction in eddy chord lengths estimated from the mooring record".*

I have a couple of other very minor editorial comments, but in general I'm quite happy with the changes made. I recommend acceptance after the comments about chord length (and the other trivial comments below) are considered.

Specific Comments

2. Lines 50-51: I think this sentence should have a reference at the end. The Thompson et al. reference (21) used for the next sentence would fit this one as well.

Response: Agreed. We have changed the text accordingly.

3. Line 142: Suggest removing "basal melt" from between "Amery Ice Shelf" and "that the mCDW".

Response: Agreed. We have changed the text accordingly.

4. Lines 211-213 and reference 54: I think it would help the reader if the authors added a little more to the text summarizing the results of the Lu and Huang study that are relevant here. Perhaps something similar to the first paragraph of their reply to Reviewer #2 in their new Response Letter?

Response: Agreed. As suggested, we have added some summarizing text and more relevant references in the manuscript as follows: *"A previous study described the eddy formation mechanism in response to large negative vorticity imparted by the passing of a storm in terms of classical Ekman layer theory⁵⁴. In the upper ocean, strong wind stress curl drives radial outward Ekman transport, resulting in an SSH decrease. Because of the continuity of mass, there must be a strong upwelling underneath the center of the storm to supply this mass flux. Assuming the frictional force is negligible, the flow is primarily cyclogeostrophic below the surface layer, and thus a cyclonic mesoscale eddy formed in the subsurface layer. There are also examples in the literature that show that wind stress curl can be a mechanism for eddy generation⁵⁵⁻⁵⁶".*

5. Lines 251-254 and Extended Data Figures 7 a+b: What are the correlation coefficients for the regressions shown in the figures?

Response: The correlation coefficients for the regressions shown in the figures 7a and 7b are 0.63 and 0.75 over 90% confidence level respectively.

6. Equation 3: Now that I understand the $T_f - T$ term better, should T_f and T be switched? As written, wouldn't this give a negative melt rate?

Response: The warm water (with T temperature) is warmer than T_f , which supplies heat flux for the melting. The negative melt rate means the volume of ice shelf is reducing. The same equation was used in Herraiz-Borrequeiro et al. (2015).

7. Figure 6 caption (also an issue in the captions for Extended Data Figures 4, 5, 6, 8, and 9): The units for wind stress curl are correct in the text for 6b, but there's a typo in the text for 6d (should be s^{-2}), not s^2).

Response: Thanks. We have changed the text accordingly.

Comment on re-revised manuscript "Persistent warm-eddy transport to Antarctic ice shelves driven by enhanced summer westerlies", with respect to the authors reply and revisions to comments from Reviewer #2. I am not Reviewer #2, but have been asked by the editor my thoughts on the authors response to those comments.

8. Reviewer comment about eddy formation mechanism: I have to admit to being

more familiar with papers either discussing the impacts of ocean eddies on atmospheric storms (especially tropical cyclones) or where wind stress weakens ocean mesoscale eddies (e.g. Xu et al., 2016 and the many references cited there), than ones where storms induce eddies in the ocean. However, there are examples in the literature where it is shown that wind stress curl can be a mechanism for eddy generation (e.g. Stammer et al. 2001, Wang et al., 2008). So, while I don't totally agree with reviewer #2 that "eddies being formed in response to large negative vorticity imparted by the passing of a storm is a new concept", I do think it's an idea that is still an open question and thus deserves more discussion in the main text than just a simple reference to the Lu and Huang paper. I mentioned in my earlier review (I am reviewer #1) that I thought it would help the reader if the authors added a little more to the text summarizing the results of the Lu and Huang study along the lines of what they put in their reply to Reviewer #2. I think perhaps a couple more references about this idea (see examples above) may also help.

Stammer et al., 2001. The role of variable wind forcing in generating eddy energy in the North Atlantic. Progress in Oceanography.

Wang et al., 2008. Winter Eddy Genesis in the Eastern South China Sea due to Orographic Wind Jets. Journal of Physical Oceanography.

Xu et al., 2016. Work done by atmospheric winds on mesoscale ocean eddies. Geophysical Research Letters.

Response: Agreed. We acknowledge that the eddy formation mechanism is still an open question and thus deserves more discussion. As the response to point 4 above, we have added some summarizing text and more relevant references in the manuscript as suggested as follows: *"A previous study described the eddy formation mechanism in response to large negative vorticity imparted by the passing of a storm in terms of classical Ekman layer theory⁵⁴. In the upper ocean, strong wind stress curl drives radial outward Ekman transport, resulting in an SSH decrease. Because of the continuity of mass, there must be a strong upwelling underneath the center of the storm to supply this mass flux. Assuming the*

frictional force is negligible, the flow is primarily cyclogeostrophic below the surface layer, and thus a cyclonic mesoscale eddy formed in the subsurface layer. There are also examples in the literature that show that wind stress curl can be a mechanism for eddy generation⁵⁵⁻⁵⁶".

9. Reviewer comment about length scale of eddy features: I did touch on something similar in my earlier comment about the chord lengths. I agree with the authors point about how "the length scales estimated from the mooring record must be smaller than the length scales seen from the SSH anomalies", but I don't think it should be a huge difference, and I think (given the assumptions I mentioned about circular eddies and an isotropic distribution) a good estimate can be made (or an argument made about why those assumptions are way off).

I can see how the length scales estimated from a mooring at 270 m could be smaller than those estimated from SSH anomalies, but I don't see how it could be "much smaller" (and the ratio could be estimated by the authors based on Figures 3d-f). I'm not sure I understand the physics involved in the authors third (eddies onshelf are smaller than offshelf eddies because of multiple meanders in the ASF?) and fourth (eddies become smaller while transiting through the Pyrdz Bay Gyre) replies to this point. I do agree with the authors that Figure 5 is illustrative of the eddy formation and intrusion onto the shelf. I still think more discussion in the manuscript is necessary in response to this point.

Response: In response to point 1 above, the chord lengths have an isotropic distribution, and even the maximum chord lengths are still an under-estimate compared to the full size of the eddies at the sea surface. As suggested, we have now added an estimation on the size of eddies over the mooring compared to the satellite estimates. We have also discussed the possible reasons for the under-estimate of eddy chords over the mooring record. The new text added in the manuscript is as follows: *"It is worth noting that the eddy chords estimated from the mooring record are around 10%~50% of the full eddy size observed from the altimetry data. There are three possible explanations for this. Firstly, the mooring record used to estimate the eddy chords is observed by a current meter located at*

270m (Figure 2b and Figure 3). Since the three-dimensional shape of eddies are vertical-taper structures (Figure 3d-f), the eddy chords at 270m must be much smaller than the length scales at the sea surface. Secondly, the majority of the eddy chords do not cross through the center of the eddies (assumed to be circular in the horizontal plane). Thirdly, the mooring is located at 67.18°S, approximately 100 km from the continental shelf break. Intruding eddies could move eastward with the Prydz Bay Gyre, preventing the entire eddy from passing over the mooring location. All three reasons explain the reduction in eddy chord lengths estimated from the mooring record".

10. Reviewer comment about the relationship between the westerlies and CDW intrusions: I think the authors have quite nicely answered reviewer 2's concerns on this point. My only suggestion is that there are several examples of modeling studies showing increasing intrusion of CDW and heat onto the Antarctic continental shelf with strengthening and poleward shifted westerlies that could be cited (perhaps starting with Spence et al, 2014).

Spence et al., 2014. Rapid subsurface warming and circulation changes of Antarctic coastal waters by poleward shifting winds. Geophysical Research Letters.

Response: Thanks for the suggestion. Spence et al. (2014) and more relevant references had been cited in the manuscript.

Point-by-point response to Reviewer #2:

Reviewer #2 (Remarks to the Author):

11. I will again start by thanking the authors for the efforts. In my last review, I raised three points where I still had significant doubts. With regards to the second and third points, on the subject of the scale of the eddies onshore/offshore and on the subject of future trends in wind stress and curl, I am happy with the response the authors have provided to me. However, only minor modifications have been made to the actual manuscript to clarify these points and they could still lead to confusion. For instance, instead of simply stating that the wind analysis is based

on daily data, the authors could be more explicit about why they think their wind stress analysis is able to capture changes in storminess. Similarly, a sentence to reconcile the order of magnitude difference between the length scales observed by the mooring and those inferred from altimetry would be useful.

Response: Thanks for the suggestion. As suggested, we have added some text to describe why the wind stress analysis is able to capture changes in storminess. The new text added in the manuscript is as follows: *"All of the associated wind-forcing analyses were based on daily wind data from the European Centre for Medium-Range Weather Forecasts (ECMWF) Reanalysis-interim (ERA-interim) products, which were able to capture both the high-frequency variability (e.g., changes in storminess) and long-term changes associated with the wind forcing"*.

As suggested, we have now added an estimation on the size of eddies over the mooring compared to the satellite estimates. We have also discussed the possible reasons for the under-estimate of eddy chords over the mooring record. The new text added in the manuscript is as follows: *"It is worth noting that the eddy chords estimated from the mooring record are around 10%~50% of the full eddy size observed from the altimetry data. There are three possible explanations for this. Firstly, the mooring record used to estimate the eddy chords is observed by a current meter located at 270m (Figure 2b and Figure 3). Since the three-dimensional shape of eddies are vertical-taper structures (Figure 3d-f), the eddy chords at 270m must be much smaller than the length scales at the sea surface. Secondly, the majority of the eddy chords do not cross through the center of the eddies (assumed to be circular in the horizontal plane). Thirdly, the mooring is located at 67.18°S, approximately 100 km from the continental shelf break. Intruding eddies could move eastward with the Prydz Bay Gyre, preventing the entire eddy from passing over the mooring location. All three reasons explain the reduction in eddy chord lengths estimated from the mooring record"*.

Moreover, we have added some text for the future changes in the Southern Hemisphere surface westerly wind stress and curl in the discussion. The new text added in the manuscript is as follows: *"Previous studies have predicted the future*

changes in the Southern Hemisphere surface westerly wind stress and curl based on observation and simulation⁵⁹⁻⁶⁰. Given the prediction that Southern Ocean westerlies will continue to strengthen and shift poleward⁵⁹⁻⁶⁰, warm-eddy intrusions onto the continental shelves are also likely to increase, with ramifications for ice shelf melting and global sea level rise".

12. With regards to the formation mechanism of the offshore eddy I strongly encourage the authors to soften the language and present it as speculation. The alternative explanation being that the observed eddies are part of the background eddy field, maintained through baroclinic instability, and can interact with the passing storms. My reasons for questioning the validity of this argument again are the following. The new reference the authors bring up in response to my comment, Lu and Huang (2010), is quite an interesting highly idealized theoretical study. As the authors of the study admit, it should be only considered as a limit, given that hurricanes are very rarely stationary (slow moving at most) and the ocean is stratified. But even then, if one was to accept the assumption of stationarity and homogenous ocean, the mechanism proposed in the study can only explain the formation of cyclones, not the anticyclones. And there is the additional problem of scales, where the scale of the oceanic eddy is an order of magnitude smaller than the scale of the storm that triggers it, when the theory predicts comparable amplitude.

Response: Agreed. As the response to point 8 above, we agree that the eddy formation mechanism is still an open question and thus deserves more discussion. We have added some summarizing text and more relevant references in the manuscript as suggested as follows: *"A previous study described the eddy formation mechanism in response to large negative vorticity imparted by the passing of a storm in terms of classical Ekman layer theory⁵⁴. In the upper ocean, strong wind stress curl drives radial outward Ekman transport, resulting in an SSH decrease. Because of the continuity of mass, there must be a strong upwelling underneath the center of the storm to supply this mass flux. Assuming the frictional force is negligible, the flow is primarily cyclogeostrophic below the*

surface layer, and thus a cyclonic mesoscale eddy formed in the subsurface layer. There are also examples in the literature that show that wind stress curl can be a mechanism for eddy generation⁵⁵⁻⁵⁶".

As suggested, we have softened the language that some of the eddy intrusion events happened without a trigger of strong wind stress curl anomaly, which might be part of the background eddy field and interacted with the passing storms. The new text added in the manuscript is as follows: *"It is worth noting that although most of the eddy intrusion events can be explained by the wind forcing, a number of the eddy events occurred without the trigger of a strong wind stress curl anomaly (Extended Data Fig. 5b). The alternative explanation is that the observed eddies are simply part of the background eddy field, maintained through baroclinic instability, and interact independently with the passing storms".*

13. Lastly advection of a subsurface feature by the Ekman flow, something the authors reiterate in their response, without further work to demonstrate it, is in my view speculation.

Response: Classical Ekman layer theory is widely used in a vast number of previous studies (e.g., Stammer et al. 2001; Wang et al., 2008; Lu and Huang, 2010; Spence et al., 2014; Xu et al., 2016), in association with wind stress, Ekman flow and air-sea interaction. The Ekman spiral and Ekman transport (Ekman flow), is a rotating column of water that forms when water moves at an angle to the wind direction due to the Coriolis Effect. The speed and direction of the moving water changes with depth, which makes a spiral to 150 meters deep. The average direction of all this turning water is about a right (left) angle from the wind direction in the Northern Hemisphere (Southern Hemisphere). So the total Ekman transport is divergence due to the cyclonic winds in the Southern Hemisphere.

Lu and Huang (2010) have demonstrated that strong cyclonic wind stress curl drives radial outward Ekman transport (divergence) in the upper ocean, resulting in an SSH decrease. There are also examples in the literature that show that wind stress curl can be a mechanism for eddy generation (e.g., Stammer et al. 2001; Wang et al., 2008).

We acknowledge that the eddy formation mechanism is still an open question and thus deserves more discussion. Further examination of this process will require more situ observations and eddy-resolution coupled modeling studies of the Southern Ocean that incorporate Antarctic ice shelves. We have now added some text in the manuscript as follows: *"The exact processes underlying the formation and transportation of eddies remain a subject of ongoing debate. Further research is essential to clarify the topical concepts presented in this study. Long-term situ observations programs around key Antarctic shelf regions are vital to effectively monitor the variability of ocean heat supply. There is also a critical need to implement eddy-resolution and associated processes into new-coupled modeling studies of the Southern Ocean that incorporate Antarctic ice shelves"*.

(1) Stammer et al., 2001. *The role of variable wind forcing in generating eddy energy in the North Atlantic. Progress in Oceanography.*

(2) Wang et al., 2008. *Winter Eddy Genesis in the Eastern South China Sea due to Orographic Wind Jets. Journal of Physical Oceanography.*

(3) Lu Z. & Huang R. *The three-dimensional steady circulation in a homogenous ocean induced by a stationary hurricane. J. Phys. Oceanogr. 40(7), 1441-1457 (2010).*

(4) Spence et al., 2014. *Rapid subsurface warming and circulation changes of Antarctic coastal waters by poleward shifting winds. Geophysical Research Letters.*

(5) Xu et al., 2016. *Work done by atmospheric winds on mesoscale ocean eddies. Geophysical Research Letters.*

REDACTED

REVIEWERS' COMMENTS

Reviewer #1 (Remarks to the Author):

Review of re-re-revised manuscript "Persistent warm-eddy transport to Antarctic ice shelves driven by enhanced summer westerlies", by Libao Gao, Xiaojun Yuan, Wenju Cai, Guijun Guo, Weidong Yu, Jiuxin Shi, Fangli Qiao, Zexun Wei, and Guy D. Williams.

General Comments

I again thank the authors for their efforts in response to my previous comments. All my previous concerns have been addressed and while I have a few new comments below, they are all very minor editorial issues. I'm happy to recommend acceptance after the comments below are considered.

Specific Comments

Lines 142-143: I think it would help the reader if they knew approximately what portion of the Amery Ice Shelf is subject to melting from mCDW. The authors give the area later in the manuscript (line 393: 12,733 km²), but I think it would help if the percentage (~12,800/62,000) was added after the word "access" on line 143 such as "(about 20% of the total area)".

Line 176: Typo, "antcyclonic" should be "anticyclonic".

Lines 189-199: Since the AVISO altimetry is provided on a 1/4 degree grid (lines 356-358), could another reason the chords from the mooring record are smaller than the eddy size estimate from the altimetry be that the smaller eddies just are not resolved by the altimetry data product?

Line 225: Since the example storm in the reference is stationary, suggest changing "passing of a storm" to "presence of a storm".

Line 231: Suggest changing "also examples" to "also other examples".

Line 233: Should "caused" be "and caused" or "causing"?

Lines 269-272, Extended Data Figures 7a+b, and Response to Reviewers #5: I thank the authors for computing the correlation coefficients and putting them in their reply. I think these values (0.63 for 7a, 0.75 for 7b) should either be added to the main text or the Extended Data Figure captions.

Line 304: Are there other alternative explanations? If so, should "The alternative" be "An alternative"?

Line 342: I think "new-coupled" should be "new coupled".

Line 385, and Response to Reviewers #6: Yes, you use the same sign convention as Herraiz-Borreguero et al. (2015). However, negative melt rate (at least to me) means freezing and increases in ice. If you're going to have negative melt rate mean that the volume of ice is decreasing, then I think it should be explicitly stated (I felt the same way about how it was used in Herraiz-Borreguero et al.).

Lines 396-401: Where are the mooring data? Apologies if I missed it, but I did not see it in any of the listed websites in this section.

Reviewer #2 (Remarks to the Author):

The authors have done a great job at responding to my comments, and am looking forward to see their manuscript published.

We have revised our manuscript accordingly with changes to the revised manuscript shown in *green italics*.

Point-by-point response to Reviewers

Reviewer #1 (Remarks to the Author):

Review of re-re-revised manuscript "Persistent warm-eddy transport to Antarctic ice shelves driven by enhanced summer westerlies", by Libao Gao, Xiaojun Yuan, WenjuCai, GuijunGuo, Weidong Yu, Jiuxin Shi, FangliQiao, Zexun Wei, and Guy D. Williams.

General Comments

I again thank the authors for their efforts in response to my previous comments. All my previous concerns have been addressed and while I have a few new comments below, they are all very minor editorial issues. I'm happy to recommend acceptance after the comments below are considered.

Specific Comments

1. Lines 142-143: I think it would help the reader if they knew approximately what portion of the Amery Ice Shelf is subject to melting from mCDW. The authors give the area later in the manuscript (line 393: 12,733 km²), but I think it would help if the percentage (~12,800/62,000) was added after the word "access" on line 143 such as "(about 20% of the total area)".

Response: Agreed. We have changed the text accordingly.

2. Line 176: Typo, "antcyclonic" should be "anticyclonic".

Response: Thanks. We have changed the text accordingly.

3. Lines 189-199: Since the AVISO altimetry is provided on a 1/4 degree grid (lines 356-358), could another reason the chords from the mooring record are smaller than the eddy size estimate from the altimetry be that the smaller eddies just are not resolved by the altimetry data product?

Response: Agreed. We have added new text in the manuscript as follows: "Finally, since the merged SSH data is provided on a 0.25° grid, the smaller eddies just are not resolved by the altimetry product".

4. Line 225: Since the example storm in the reference is stationary, suggest changing

"passing of a storm" to "presence of a storm".

Response: Agreed. We have changed the text accordingly.

5. Line 231: Suggest changing "also examples" to "also other examples".

Response: Agreed. We have changed the text accordingly.

6. Line 233: Should "caused" be "and caused" or "causing"?

Response: Agreed. We use "and caused" here.

7. Lines 269-272, Extended Data Figures 7a+b, and Response to Reviewers #5: I thank the authors for computing the correlation coefficients and putting them in their reply. I think these values (0.63 for 7a, 0.75 for 7b) should either be added to the main text or the Extended Data Figure captions.

Response: The Extended Data Figures 7a+b show the diagrams of warm signals to zonal wind and wind stress curl. We just plot two regression lines in the plots, so I think it is not necessary to add the correlation coefficients to the main text. Moreover, the readers can get the information from the rebuttal files.

8. Line 304: Are there other alternative explanations? If so, should "The alternative" be "An alternative"?

Response: Agreed. We have changed the text accordingly.

9. Line 342: I think "new-coupled" should be "new coupled".

Response: Agreed. We have changed the text accordingly.

10. Line 385, and Response to Reviewers #6: Yes, you use the same sign convention as Herraiz-Borreguero et al. (2015). However, negative melt rate (at least to me) means freezing and increases in ice. If you're going to have negative melt rate mean that the volume of ice is decreasing, then I think it should be explicitly stated (I felt the same way about how it was used in Herraiz-Borreguero et al.).

Response: Agreed. We have added new text in the manuscript as follows:

"Negative melt rate means the volume of ice is decreasing".

11. Lines 396-401: Where are the mooring data? Apologies if I missed it, but I did not see it in any of the listed websites in this section.

Response: We have updated the data link which can find the data directly.

Reviewer #2 (Remarks to the Author):

The authors have done a great job at responding to my comments, and am looking forward to see their manuscript published.

Response: Thanks.